# On Efficiency-Effectiveness Trade-off of Diffusion-based Recommenders

**Wenyu Mao[1], Jiancan Wu[1,2]∗, Guoqing Hu[1], Zhengyi Yang[1], Wei Ji[3], Xiang Wang[1]∗,**
[1] University of Science and Technology of China
[2]Institute of Dataspace, Hefei Comprehensive National Science Center
[3] Nanjing University

## Abstract

Diffusion models have emerged as a powerful paradigm for generative sequential recommendation, which typically generate next items to recommend guided by user interaction histories with a multi-step denoising process. However, the multi-step process relies on discrete approximations, introducing discretization error that creates a trade-off between computational efficiency and recommendation effectiveness. To address this trade-off, we propose TA-Rec, a two-stage framework that achieves one-step generation by smoothing the denoising function during pretraining while alleviating trajectory deviation by aligning with user preferences during fine-tuning. Specifically, to improve the efficiency without sacrificing the recommendation performance, TA-Rec pretrains the denoising model with Temporal Consistency Regularization (TCR), enforcing the consistency between the denoising results across adjacent steps. Thus, we can smooth the denoising function to map the noise as oracle items in one step with bounded error. To further enhance effectiveness, TA-Rec introduces Adaptive Preference Alignment (APA) that aligns the denoising process with user preference adaptively based on preference pair similarity and timesteps. Extensive experiments prove that TA-Rec's two-stage objective effectively mitigates the discretization errors-induced trade-off, enhancing both efficiency and effectiveness of diffusion-based recommenders. Our code is available at https://github.com/maowenyu-11/TA-Rec.

## 1 Introduction

Diffusion models [1–3] have recently shown strong potential in generative sequential recommendation [4–6, 4, 7], owing to their remarkable ability to model complex distributions of user behaviors and generate oracle items that best match user preferences. At the core of diffusion-based sequential recommendation [4, 5, 7] is framing a theoretical continuous process [2, 3] which gradually reconstructs the oracle items from random noise. Typically, such theoretical continuous process (*e.g.,* Stochastic Differential Equations (SDE) [2]) is realized by multi-step discrete approximations [8–11] (*e.g.,* Denoising Diffusion Probabilistic Models (DDPM) [1]) for practical implementation: they add noise to the ground-truth next items in the forward process and then denoise it step by step in the reverse process with denoising models, guided by the interaction history [12].

While achieving success, this discrete approximation inherently introduces discretization errors [13] between the practical reverse trajectory (*i.e.,* the dashed line of discrete DDPM in Figure 1a) and the theoretical trajectory (*i.e.,* the solid line of continuous SDE in Figure 1a). Such discretization error is primarily caused by high-order truncation in numerical solvers and accumulates across the multi-step reverse process. When approximating SDE [2] with finite steps $T$, discretization errors grow as $T$ decreases [14, 15], creating an efficiency-effectiveness trade-off: increasing the number of steps $T$

---

∗Corresponding author: wujcan@gmail.com, xiangwang@ustc.edu.cn.

39th Conference on Neural Information Processing Systems (NeurIPS 2025).

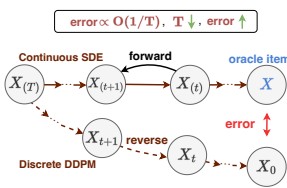

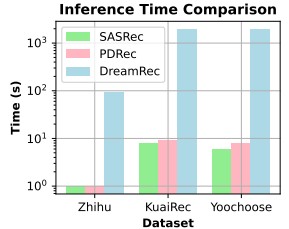

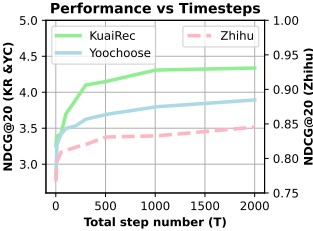

(a) The discretization error.    (b) Inference time comparison.    (c) Effectiveness and efficiency.

Figure 1: Overall motivation for our work, including the existence of discretization error, high inference costs for diffusion-based recommenders, and the effectiveness and efficiency trade-off.

(typically 1,000 steps) for diffusion-based recommenders improves alignment between generated items and user preferences but incurs substantial inference overhead, while aggressively reducing total steps $T$ accelerates generation at the cost of amplified discretization error and misalignment. As shown in Figure 1b, compared with traditional recommenders (*e.g.,* SASRec [16] and PDRec [17]), the diffusion-based DreamRec [4] requires 1,000 steps to achieve comparable performance, significantly slowing inference. Conversely, reducing $T$ significantly boosts acceleration yet leads to notable performance drops of DreamRec — evidenced by a significant NDCG drop from $T = 2,000$ to $T = 1$ on different datasets in Figure 1c.

To resolve the trade-off, we propose TA-Rec, a two-stage framework that realizes one-step generation by smoothing the denoising function in pretraining stage and reduces the trajectory deviation by optimizing denoising models with user preferences in fine-tuning stage. Technically, in the pretraining stage, TA-Rec proposes temporal consistency regularization (TCR) to enforce consistency between denoising representations at adjacent timesteps, smoothing the denoising function [18] to map noise to oracle items directly [15, 14] with bounded error. This could improve diffusion models' efficiency in one-step generation without compromising the recommendation performance. In the fine-tuning stage, TA-Rec introduces adaptive preference alignment (APA) [7, 19] to mitigate the trajectory deviation and enhance recommenders' effectiveness. Specifically, APA constructs preference pairs by selecting positive and negative items from the interaction space. Recognizing that the denoising trajectory from noise to oracle items inherently varies with both user preferences and noise magnitude, APA dynamically calibrates alignment strength according to both the similarity between preference pairs and the time steps of the noisy input, optimizing denoising models in a more refined manner. Notably, when the preference pair is hard to distinguish and the noisy input has a large noise degree, APA strategically reduces optimization strength to avoid overfitting to ambiguous preferences and random noise.

Theoretical analysis demonstrates that TA-Rec has bounded error despite accelerated generation and achieves more accurate alignment, thereby enabling reliable recommendations matching user preference. Empirical results across multiple real-world benchmarks (*i.e.,* zhihu [20], KuaiRec [21], and YooChoose [22]) show that our approach achieves $100\times$ faster generation compared to leading diffusion-based recommenders (*e.g.,* DreamRec [4]) while simultaneously improving recommendation performance by $10\%$, achieving both higher efficiency and higher effectiveness.

## 2  Realated Wrok

**Diffusion-based Recommenders** [4, 6, 23–25] have emerged as a promising alternative to conventional sequential recommenders by modeling the complex user behavior distribution and generating next items step by step to match user preference. Existing methods have explored various applications of diffusion models in recommendation systems, including next-item generation through a denoising process guided by historical sequences [4, 5, 23], augmentation of sequential recommenders [26, 17, 27], and improvement of robustness [28] against noisy feedback [29, 30]. Our work addresses the issue of discretization error [13] in the reverse process for next-item generation paradigm, which fundamentally impacts the efficiency-effectiveness trade-off that currently limits diffusion-based recommenders' practical deployment.

**Accelerating Diffusion** [14, 31–34] addresses the computational bottleneck inherent in the multi-step generation process of diffusion models. Recent advancements fall into two primary categories: training-free numerical solvers [14, 35, 36] and training-based model distillation [37–40]. The former

optimizes numerical solvers to serve as faster samplers during the reverse process, reducing inference steps to 20-50 without tuning the original denoising model [14, 41, 42]. For instance, DPM++ [43] leverages adaptive high-order solvers to approximate the continuous SDEs [2], achieving faster generation via optimized numerical integration. To achieve more aggressive acceleration (1-4 steps), the latter approach focuses on tuning the denoising model through knowledge distillation. Specifically, consistency models [15, 44] distill pre-trained diffusion models into single-step generators by enforcing self-consistency of the ODE trajectory. However, these methods may either rely on well-pretrained models or a deterministic ODE process, limiting their applicability to recommendation tasks due to the lack of unified recommendation benchmarks and the dynamic nature of user preferences. Instead, our work achieves one-step generation of diffusion-based recommenders by designing a temporal consistency loss for the stochastic DDPM process.

**Aligning Diffusion Models** [19, 45–47] aim to optimize diffusion models using human preference data, an area that remains relatively underdeveloped compared to preference optimization in large language models (LLMs). Current approaches often adapt existing LLM preference optimization algorithms to diffusion models. For example, Diffusion-DPO [19] takes the DPO loss [48] from LLMs and adapts it on diffusion models directly to improve image generation based on user preference. DSPO [47] aligns diffusion models with human preferences by distilling the score function of preferred image distributions into the model's pretrained score functions, leveraging score matching. Furthermore, Diffusion-NPO [46] approaches preference alignment by training an additional model to specifically model negative preferences. To enable alignment at a more fine-grained level, our work designs an adaptive preference optimization algorithm that aligns the generation of diffusion-based recommenders according to both timesteps and pair similarity.

## 3 Preliminaries

### 3.1 Diffusion-based Sequential Recommendation

In sequential recommendation tasks, we represent a user's historical interaction sequence chronologically as $\mathbf{x}^{1:N-1} = [\mathbf{x}^1, \mathbf{x}^2, \ldots, \mathbf{x}^{N-1}]$ in the embedding space, where each $\mathbf{x}^n \in \mathbb{R}^d$ denotes the embedding of the user's $n$-th interacted item. The next item to be interacted with is denoted as $\mathbf{x}^N$ ($\mathbf{x}$ for simplification). The goal of diffusion-based sequential recommendation is to recover the oracle item [4] that best aligns with users' preferences from noise, guided by user interaction history. Following [4, 23, 7], in the forward process, Gaussian noise is added to the next item $\mathbf{x}$: $q(\mathbf{x}_t|\mathbf{x}) = \mathcal{N}(\mathbf{x}_t; \sqrt{\bar{\alpha}_t}\mathbf{x}, (1 - \bar{\alpha}_t)\mathbf{I})$, where $[\alpha_1, \ldots, \alpha_T]$ denotes the noise scale, $t \in [1, \ldots, T]$ denotes the timestep. During the reverse process, diffusion models recover the next item $\mathbf{x}$ from noise under the guidance $\mathbf{g}$ through a multi-step denoising process. To optimize the denoising model, the training loss for diffusion-based recommenders can be formulated as [4, 7]:

$$\mathcal{L}_{\text{diff}} = \mathbb{E}_{\mathbf{x},\mathbf{g},t} \left[ \|f_\theta(\mathbf{x}_t, \mathbf{g}, t) - \mathbf{x}\|_2^2 \right], \tag{1}$$

where $\mathbf{g}$ is the guidance signal extracted from users' historical interaction sequences $\mathbf{x}^{1:N-1}$ with a Transformer following [4, 7], $f_\theta(\cdot)$ is a denoising model parameterized by MLP, **reconstructing** the target item $\mathbf{x}$ as $\hat{\mathbf{x}}_0$ directly under the guidance $\mathbf{g}$, as shown in the green part of Figure 2. Commonly, classifier-free guidance paradigm [12] can be utilized by replacing the guidance $\mathbf{g}$ with a dummy token $\Phi$ at probability $\rho$, allowing for the integration of conditional and unconditional training. During inference, the pure Gaussian noise $\mathbf{x}_T \sim \mathcal{N}(0, \mathbf{I})$ serves as the input, then the denoising model $f_\theta(\cdot)$ denoises items as $\hat{\mathbf{x}}_0$ under the guidance $\mathbf{g}$. The iterative reverse process for generation is:

$$\mathbf{x}_{t-1} = \frac{\sqrt{\bar{\alpha}_{t-1}}\beta_t}{1 - \bar{\alpha}_t}\hat{\mathbf{x}}_0 + \frac{\sqrt{\alpha_t}(1 - \bar{\alpha}_{t-1})}{1 - \bar{\alpha}_t}\mathbf{x}_t + \sqrt{\tilde{\beta}_t}\mathbf{z}, \quad \mathbf{z} \sim \mathcal{N}(\mathbf{0}, \mathbf{I}). \tag{2}$$

After generating the oracle item $\mathbf{x}_0$ step-by-step, we calculate the dot product between $\mathbf{x}_0$ and each candidate item in the corpus, then recommend $K$ items having the highest similarity scores.

### 3.2 Direct Preference Optimization on Diffusion Models

Direct Preference Optimization (DPO) [49–51] is a reward-model-free method that aligns LLMs with human preferences via a supervised loss derived from pairwise comparison. To adapt DPO [48] to the diffusion model [19, 47, 46], the key idea is to increase denoising models' likelihood of preferred

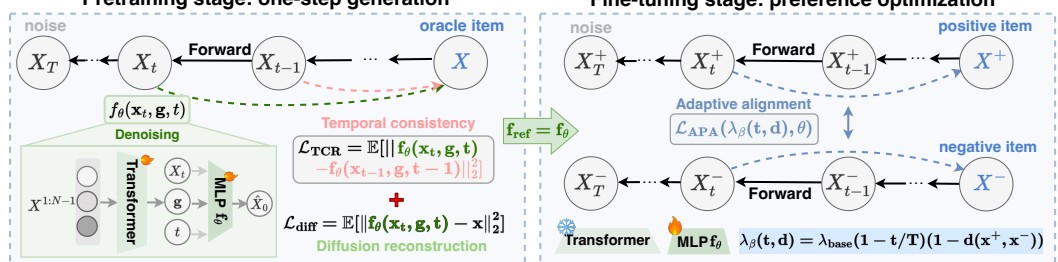

Figure 2: The overview of our proposed TA-Rec framework, which alleviates the efficiency and effectiveness trade-off of diffusion-based recommenders with two stages. In the pretraining stage, TA-Rec achieves one-step generation with temporal consistency regularization. In the fine-tuning stage, TA-Rec adopts an adaptive coefficient $\lambda_\beta(t, d)$ to align the generated item with users' preferences.

generation while decreasing that of dispreferred ones. We assume the pairwise preference data as $D = (\mathbf{x}^+, \mathbf{x}^-)$, where $\mathbf{x}^+$ is preferred over $\mathbf{x}^-$. Following [19], the DPO training loss on diffusion model can be formulated as:

$$\mathcal{L}_{\text{Diffusion-DPO}} = -\mathbb{E}\left[\log \sigma\left(\lambda_\beta \log \frac{p_\theta\left(\mathbf{x}_t^+ \mid \mathbf{x}_{t+1}^+, \mathbf{g}\right)}{p_{\text{ref}}\left(\mathbf{x}_t^+ \mid \mathbf{x}_{t+1}^+, \mathbf{g}\right)} - \lambda_\beta \log \frac{p_\theta\left(\mathbf{x}_t^- \mid \mathbf{x}_{t+1}^-, \mathbf{g}\right)}{p_{\text{ref}}\left(\mathbf{x}_t^- \mid \mathbf{x}_{t+1}^-, \mathbf{g}\right)}\right)\right], \quad (3)$$

where $\mathbf{x}_t^+ \sim q(\mathbf{x}_t^+ \mid \mathbf{x}_0^+), \mathbf{x}_t^- \sim q(\mathbf{x}_t^+ \mid \mathbf{x}_0^+)$, $\mathbf{g}$ is the guidance of denoising model, $p_{\text{ref}}$ represents the reference distribution, $\lambda_\beta$ is a hyperparameter which controls preference optimization strength.

## 4 Method

In this section, we present our proposed TA-Rec framework, a two-stage approach addressing the discretization error-induced trade-off in diffusion-based recommenders. The overall framework is illustrated in Figure 2. We first define the discretization error of diffusion models in Section 4.1. We then detail the Temporal Consistency Regularization (TCR) during pretraining in Section 4.2. Finally, we describe the Adaptive Preference Alignment (APA) during fine-tuning in Section 4.3.

### 4.1 Definition of Discretization Error in Diffusion Models

The continuous reverse process of the diffusion model can be described by the SDE process [2] mathematically. Formally, we have: $d\mathbf{x} = \left[f(\mathbf{x}, s) - g(s)^2 \nabla_\mathbf{x} \log p_s(\mathbf{x})\right] ds + g(s) d\bar{w}$, where $f(\cdot)$ and $g(\cdot)$ are drift and diffusion coefficients, $\bar{w}$ is reverse-time Brownian motion, and $p_s(\mathbf{x})$ is the marginal distribution at time $s$ ($s \in [0, 1]$). In practice, the theoretical continuous SDE is approximated by discrete numerical solvers (*e.g.,* Euler-Maruyama [52], a first-order solver) with $T$ steps (step size $\Delta t = 1/T$): $\mathbf{x}_{t+1} = \mathbf{x}_t + \left[f - g^2 \nabla \log p_t\right] \Delta t + g\sqrt{\Delta t}\mathbf{z}$, $\mathbf{z} \sim \mathcal{N}(0, \mathbf{I})$.

**Definition 1** *The deviation between the continuous process and its discrete approximation is the discretization error $\mathcal{E}_{disc}$ : $\mathcal{E}_{\text{disc}}(t) = \mathbb{E}_{\mathbf{x} \sim p_{(t)}}\left[\left\|\mathbf{x}_{(t)} - \mathbf{x}_t\right\|_2\right]$, where $\mathbf{x}_{(t)}$ is the theoretical SDE solution at time $s = t \cdot \Delta t$, $\mathbf{x}_t$ is the discrete approximation at step $t$, $t \in [1, \ldots, T]$.*

Such discretization error primarily stems from local truncation error in solvers' numerical integration [53, 54] and is accumulated globally during the propagation across multiple steps in the reverse process [55, 56], where $\mathcal{E}_{\text{disc}}(0) \sim O(\Delta t)$. Thus, reducing reverse steps $T$ (*i.e.,* increasing $\Delta t$) will amplify $\mathcal{E}_{\text{disc}}$ and cause trajectory deviation, introducing the efficiency and effectiveness trade-off.

### 4.2 Pretraining Stage: Temporal Consistency Regularization (TCR)

In recommendation systems, the multi-step reverse process introduces high computational costs, failing to meet the low-latency demands of large-scale online deployments. To improve the efficiency of diffusion-based recommenders, we aim to realize one-step generation, which maps the noise to the oracle items directly. Specifically, TA-Rec proposes temporal consistency regularization (TCR) to smooth the denoising function of diffusion-based recommenders in the pertaining stage, as shown

in the left part of Figure 2. In addition to the reconstruction loss in Equation (1), TCR enforces consistency between consecutive timesteps to smooth the denoising function $f_\theta(\cdot)$ with Lipschitz continuity [18]. For adjacent noisy representations $\mathbf{x}_t$ and $\mathbf{x}_{t-1}$ (obtained via the forward process $q(\mathbf{x}_t|\mathbf{x}) = \mathcal{N}(\mathbf{x}_t; \sqrt{\bar{\alpha}_t}\mathbf{x}, (1 - \bar{\alpha}_t)\mathbf{I})$ as introduced in Section 3.1), the TCR loss can be defined as:

$$\mathcal{L}_{\text{TCR}} = \mathbb{E}_{\mathbf{x}_t, \mathbf{g}, t} \left[ \|f_\theta(\mathbf{x}_t, \mathbf{g}, t) - f_\theta(\mathbf{x}_{t-1}, \mathbf{g}, t-1)\|_2^2 \right], \tag{4}$$

where $\mathbf{g}$ is the guidance signal extracted from the historical interaction sequence $\mathbf{x}^{1:N-1}$ with a Transformer, as introduced in Section 3.1. The total loss, which jointly optimizes the denosing model with reconstruction loss in Equation (1) and TCR loss in Equation (4), can be formulated as:

$$\mathcal{L}_{\text{pre}} = \mathcal{L}_{\text{diff}} + \lambda_c \cdot \mathcal{L}_{\text{TCR}}, \tag{5}$$

where $\lambda_c$ is the strength that balances reconstruction accuracy and denoising smoothness. Upon optimizing $\mathcal{L}_{\text{pre}}$, we can obtain the smooth denoising function $f_\theta(\mathbf{x}_t, \mathbf{g}, t)$ that consistently projects the noisy representation $\mathbf{x}_t$ at any step on the trajectory to the target item $\mathbf{x}$. Thus, the pure noise $\mathbf{x}_T \sim \mathcal{N}(0, \mathbf{I})$ can be mapped by the smooth denoising function as oracle items $\mathbf{x}_0$ directly with one-step generation, eliminating the need for iterative reverse steps. Below, we justify that such accelerated generation has a bounded error.

**Theorem 1** *(Error Bound for One-Step Generation) We assume that: (i) The smooth denoising function $f_\theta(\mathbf{x}_t, \mathbf{g}, t)$ satisfies Lipschitz continuity with constants $L > 0$: $\|f_\theta(\mathbf{x}_{s_1/\Delta t}, \mathbf{g}, s_1/\Delta t) - f_\theta(\mathbf{x}_{s_2/\Delta t}, \mathbf{g}, s_2/\Delta t)\|_2 \leq L|s_1 - s_2|$ for any time $s_1, s_2 \in [0, 1]$. (ii) For any step $t \in [1, \ldots, T]$ and any time $s \in [t\Delta t, (t+1)\Delta t]$, there exists a constant $C$ such that the following smooth condition holds $\|f_\theta(\mathbf{x}_{s/\Delta t}, \mathbf{g}, s/\Delta t) - f_\theta(\mathbf{x}_t, \mathbf{g}, t)\|_2 \leq C\|f_\theta(\mathbf{x}_{t+1}, \mathbf{g}, t+1) - f_\theta(\mathbf{x}_t, \mathbf{g}, t)\|_2$. Then, since we minimize $L_{pre}$, the error of one-step generation of $\mathbf{x}$ from any $s$ is bounded as follows:*

$$\sup_{s, \mathbf{x}} \|f_\theta(\mathbf{x}_{s/\Delta t}, \mathbf{g}, s/\Delta t) - \mathbf{x}\|_2 = O\left(\Delta t\right). \tag{6}$$

Thus, the error of one-step generation remains bounded by the global discretization error $O(\Delta t)$ in multi-step reverse process, validating that acceleration will not amplify trajectory deviation or disrupt the recommenders' effectiveness. The full proof is provided in Appendix A.1.

### 4.3 Fine-tuning Stage: Adaptive Preference Alignment (APA)

After improving the efficiency as detailed in Section 4.2, we further enhance diffusion-based recommenders' effectiveness by aligning the generated items with users' preferences closely, mitigating the trajectory deviation caused by the discretization error. Technically, TA-Rec introduces adaptive preference alignment (APA) in the fine-tuning stage to optimize the denoising model $f_\theta(\cdot)$ with users' preference pairs. To align the denoising trajectory with user preferences in a more refined manner, APA designs an adaptive coefficient $\lambda_\beta$, which controls the strength of preference optimization (as introduced in Section 3.2) adaptively. For each sequence $\mathbf{x}^{1:N-1}$, we randomly sample the negative item $\mathbf{x}^-$ from the batch, constructing the preference pair with ground-truth next item $\mathbf{x}^+$ (*i.e.,* $\mathbf{x}$). Since denoising from noise to oracle item is relative to user preference and the noisy degree of $\mathbf{x}_t$, the optimization strength $\lambda_\beta$ can adapt at both pair- and step-wise. For a preference pair $(\mathbf{x}^+, \mathbf{x}^-)$, we define the similarity between positive and negative items as $d(\mathbf{x}^+, \mathbf{x}^-) = \text{cosine}(\mathbf{x}^+, \mathbf{x}^-)$, where cosine denotes the cosine similarity. Thus, the alignment strength $\lambda_\beta(t, s)$ can adapt according to timestep $t$ and pair similarity $d(\mathbf{x}^+, \mathbf{x}^-)$:

$$\lambda_\beta(t, d) = \lambda_{\text{base}} \cdot \left( \left(1 - \frac{t}{T}\right) + \left(1 - d(\mathbf{x}^+, \mathbf{x}^-)\right) \right), \tag{7}$$

where $\lambda_{\text{base}}$ controls the base alignment strength, we set it as $1/2$. When the pair similarity $d(\mathbf{x}^+, \mathbf{x}^-)$ and the step $t$ is large (*i.e.,* the preference pair is hard to distinguish and $\mathbf{x}_t$ has a large noisy degree), the optimization strength $\lambda_\beta(t, d)$ can be small, which can avoid overfitting to ambiguous preferences and random noise. The APA loss integrates this adaptive strength $\lambda_\beta(t, d)$ into the DPO loss of Diffusion as introduced in Equation (3), and we have:

$$\mathcal{L}_{\text{APA}} = -\mathbb{E}\left[ \log \sigma \left( \lambda_\beta(t, d) \log \frac{p_\theta\left(\mathbf{x}_t^+ \mid \mathbf{x}_{t+1}^+, \mathbf{g}\right)}{p_{\text{ref}}\left(\mathbf{x}_t^+ \mid \mathbf{x}_{t+1}^+, \mathbf{g}\right)} - \lambda_\beta(t, d) \log \frac{p_\theta\left(\mathbf{x}_t^- \mid \mathbf{x}_{t+1}^-, \mathbf{g}\right)}{p_{\text{ref}}\left(\mathbf{x}_t^- \mid \mathbf{x}_{t+1}^-, \mathbf{g}\right)} \right) \right], \tag{8}$$

where $\mathbf{x}_t^+$ and $\mathbf{x}_t^-$ are noisy representations of $\mathbf{x}^+$ and $\mathbf{x}^-$ (obtained by $q(\mathbf{x}_t|\mathbf{x}) = \mathcal{N}(\mathbf{x}_t; \sqrt{\bar{\alpha}_t}\mathbf{x}, (1-\bar{\alpha}_t)\mathbf{I}))$, $p_{\text{ref}}$ denotes the reference distribution, which can be initialized using the distribution from the pretrained denoising model. The Transformer model to extract guidance signals $\mathbf{g}$ is frozen in the fine-tuning stage. Similar to [19], the loss to optimize denoising model $f_\theta(\cdot)$ can be simplified as:

$$
\begin{aligned}
\mathcal{L}_{\text{APA}} = -\mathbb{E}\Big[ & \log\sigma\Big(-\lambda_\beta(t,d)\Big(\big\|f_\theta(\mathbf{x}_t^+, \mathbf{g}, t) - \mathbf{x}^+\big\|_2^2 - \big\|f_{\text{ref}}(\mathbf{x}_t^+, \mathbf{g}, t) - \mathbf{x}^+\big\|_2^2 \\
& -\Big(\big\|f_\theta\left(\mathbf{x}_t^-, \mathbf{g}, t\right) - \mathbf{x}^-\big\|_2^2 - \big\|f_{\text{ref}}\left(\mathbf{x}_t^-, \mathbf{g}, t\right) - \mathbf{x}^-\big\|_2^2\Big)\Big)\Big)\Big],
\end{aligned}
\tag{9}
$$

where $f_{\text{ref}}(\cdot)$ is the reference model initialized by the denoising model from the pertaining stage. Below, we justify that the pair- and step-aware adaptive coefficient $\lambda_\beta(t, d)$ allows the denoising model to align the generation with user preferences more closely.

**Theorem 2** *Suppose that the preference of $f_\theta$ and $f_{ref}$ have the following relation: $f_{ref}(\mathbf{x}_t^+, \mathbf{g}, t)/f_{ref}(\mathbf{x}_t^-, \mathbf{g}, t) \leq f_\theta(\mathbf{x}_t^+, \mathbf{g}, t)/f_\theta(\mathbf{x}_t^-, \mathbf{g}, t) \leq k f_{ref}(\mathbf{x}_t^+, \mathbf{g}, t)/f_{ref}(\mathbf{x}_t^-, \mathbf{g}, t)$, where $k$ is a constant and $k \leq e$. Then with increasing $\lambda_\beta$, the parameter update of $\theta$ from preference optimization becomes more aggressive, i.e., the norm of gradient $\|\nabla_\theta \mathcal{L}_{Diffusion\text{-}DPO}\| \propto \lambda_\beta$.*

Theorem 2 justifies that the denoising model's parameter $\theta$ updates more slowly when the coefficient $\lambda_\beta(t, d)$ is small. The proof is detailed in the Appendix A.2. When the pair similarity $d(\mathbf{x}^+, \mathbf{x}^-)$ and the step $t$ is large (*i.e.,* the preference pair is hard to distinguish and $\mathbf{x}_t$ has a large noisy degree), we can have a small optimization strength $\lambda_\beta(t, d)$. Thus, parameters $\theta$ of the denoising model can update more slowly, which can avoid overfitting to ambiguous preferences and random noise. And vice versa. Such adaptive strategy allows the denoising model to align the generation with user preferences more closely in a more refined manner.

## 4.4 Overall Pipeline

As shown in Figure 2, the pipeline of TA-Rec begins with pretraining the denoising model under Temporal Consistency Regularization (TCR) to enable efficient one-step generation. Given a user's interaction sequence $\mathbf{x}^{1:N-1}$, we first extract guidance signals $g$ using a Transformer encoder. The adjacent noisy representations $\mathbf{x}_t$ and $\mathbf{x}_{t-1}$ are generated via the forward process $q(\mathbf{x}_t|\mathbf{x}) = \mathcal{N}(\mathbf{x}_t; \sqrt{\bar{\alpha}_t}\mathbf{x}, (1-\bar{\alpha}_t)\mathbf{I})$, where $\mathbf{x}$ is the ground-truth next item. The Transformer and denoising model is then optimized using a joint loss $\mathcal{L}_{\text{pre}} = \mathcal{L}_{\text{diff}} + \lambda_c \cdot \mathcal{L}_{\text{TCR}}$. In the fine-tuning stage, Adaptive Preference Alignment (APA) fine-tunes the denosing model $f_\theta(\cdot)$ to better align generated items with user preference. For each sequence, we construct preference pairs $(\mathbf{x}^+, \mathbf{x}^-)$ by randomly sampling items that users have not interacted with as negatives. The alignment strength $\lambda_\beta(t, d)$ is dynamically adapted based on both the timestep $t$ and pair similarity $d(\mathbf{x}^+, \mathbf{x}^-)$ as detailed in Equation (7). The APA loss $\mathcal{L}_{\text{APA}}$ optimizes the denoising model with the guidance encoder frozen.

During inference, the model generates recommendations in a single step: given a user's history $\mathbf{x}^{1:N-1}$, we obtain guidance $g$ and sample noise $\mathbf{x}_T \sim \mathcal{N}(0, \mathbf{I})$ as input. Then, we generate oracle items with the denoising model: $\mathbf{x}_0 = f_\theta(\mathbf{x}_T, \mathbf{g}, T)$. The final recommendations are obtained by ranking candidate items via dot product with generated items $\mathbf{x}_0$. The algorithm of the pertaining stage, fine-tuning stage, and the inference phase of TA-Rec are presented in Appendix B.

## 5 Experiments

In this section, we conduct extensive experiments to evaluate how TA-Rec addresses the trade-off by answering the following questions: RQ1: How does TA-Rec perform in the sequential recommendation tasks compared with leading baselines? RQ2: What are the contributions of TCR and APA in TA-Rec? RQ3: How sensitive is TA-Rec to the strength of consistency regularization and preference optimization? RQ4: How efficient is TA-Rec compared to traditional recommenders and diffusion-based recommenders? RQ5: Can TA-Rec generalize to multi-step reverse process and different pretrained diffusion-based recommenders?

### 5.1 Expermental Settings

**Datasets.** We adopt three common datasets in sequential recommendation tasks to conduct the experiments, including Yoochoose [22], KuaiRec [21], and Zhihu [20]. To process the dataset, we

exclude items with fewer than five interactions and sequences shorter than 3 interactions to mitigate cold-start issues following [4]. Then, we sort all sequences chronologically and split the data into training, validation, and testing sets in an 8:1:1 ratio to prevent data leakage. The dataset statistics are provided in Appendix C.1.

**Baselines.** We evaluate the performance of TA-Rec against multiple leading sequential recommenders thoroughly, including:

- Traditional Recommender: GRU4Rec [57], Caser [58], SASRec [16], Bert4Rec [59], and CL4SRec [60] predict next items by calculating the similarity between candidate items and the interaction sequences, which are modeled using GRU and Transformer architectures.

- Diffusion-based Recommender: DiffRec [6], DiffuRec [5], DreamRec [4] leverage diffusion models to formulate the adding noise and denoising process in recommenders, generating item embeddings or item scores step by step.

- Preference-based Recommender: DiffuASR [26], PDRec [17], DimeRec [61], PreferDiff [7]. DiffuASR [26], PDRec [17] employ diffusion models to generate augmented items that enrich user preference representations. Meanwhile, DimeRec [61] and PreferDiff [7] directly generate recommended items using diffusion models, enhanced by multi-interest extraction and preference pair optimization, respectively.

The detailed explanation for baselines is presented in the Appendix C.2

**Implementation Details.** Following DreamRec [4], the historical interaction sequence length is set to 10, with sequences containing fewer than 10 interactions padded using a padding token. Item embeddings are dimensioned at 256 for the Zhihu dataset and 64 for the KuaiRec and Yoochoose datasets. The learning rate during the pretraining stage is tuned within the range of $[0.01, 0.005, 0.001, 0.0005, 0.0001, 0.00005]$. The timesteps $T$ for forward process are varied across $[500, 1,000, 2,000]$. The hyperparameter of $\lambda_c$ is tuned across the range $[0.1, \ldots, 1]$. The experiments are implemented with Python 3.9 and PyTorch 2.0.1 on the Nvidia GeForce RTX 3090. We employ widely used metrics in sequential recommendation: hit ratio (HR@20) and normalized discounted cumulative gain (NDCG@20) for evaluation. Each method is tested five times, with the average performance and corresponding standard deviations reported in the tables.

## 5.2 Main Results (RQ1)

To answer RQ1 and validate the effectiveness of TA-Rec, we present the overall recommendation performance on three datasets in Table 1. To implement, all the methods that leverage diffusion models are based on DDPM [1] with multiple reverse steps, while PreferDiff is based on DDIM [35] for acceleration (10-20 steps) according to the original setting in [7]. Our proposed TA-Rec consistently outperforms leading baselines of three categories across all datasets, demonstrating its superiority in recommendation effectiveness. As shown in Table 1, TA-Rec achieves the best performance under both HR@20 and NDCG@20 metrics, with significant improvements over the strongest baselines (7.14%-31.82% on YooChoose, 4.65%-7.54% on KuaiRec, and 7.52%-9.64% on Zhihu). DreamRec [4] demonstrates competitive performance but incurs high computational costs due to its 1,000-step generation process. PreferDiff [7] demonstrates excellent performance by integrating BPR loss with diffusion reconstruction loss and enabling faster generation (10-20 steps) through the adoption of DDIM; however, it still trails behind TA-Rec. The superiority of our method with one-step generation validates the success of our two-stage framework in enhancing both the efficiency and effectiveness of diffusion-based recommenders.

## 5.3 Abalation Study (RQ2)

To validate the respective contribution of TCR and APA in TA-Rec, we conduct ablation studies with experimental results presented in Table 2. Specifically, we design several variants for TCR, where "DDPM" represents adopting the DDPM [1] paradigm to generate items step-by-step (1k steps) to recommend, guided by user interaction history. "DDIM" refers to leveraging DDIM [35] to accelerate the multi-step reverse process to just 10-20 steps. "w/o TCR" denotes fine-tuning denosing models from pretrained DDPM without TCR loss, "w/o APA" denotes pretraining denosing models without the fine-tuning stage. "w/o t", "w/o d", "w/o td" represent conduct preference alignment without

Table 1: Overall performance of different methods of sequential recommendation. The best score and the second-best score are bolded and underlined, respectively. The last row indicates the performance improvements of TA-Rec over the best-performing baseline method.

| Methods | YooChoose | | KuaiRec | | Zhihu | |
|---|---|---|---|---|---|---|
| | HR@20 | NDCG@20 | HR@20 | NDCG@20 | HR@20 | NDCG@20 |
| GRU4Rec | $3.89_{\pm 0.11}$ | $1.62_{\pm 0.02}$ | $3.32_{\pm 0.11}$ | $1.23_{\pm 0.08}$ | $1.78_{\pm 0.12}$ | $0.67_{\pm 0.03}$ |
| Caser | $4.06_{\pm 0.12}$ | $1.88_{\pm 0.09}$ | $2.88_{\pm 0.19}$ | $1.07_{\pm 0.07}$ | $1.57_{\pm 0.05}$ | $0.59_{\pm 0.01}$ |
| SASRec | $3.79_{\pm 0.03}$ | $1.71_{\pm 0.03}$ | $4.02_{\pm 0.09}$ | $1.79_{\pm 0.10}$ | $1.85_{\pm 0.01}$ | $0.77_{\pm 0.03}$ |
| Bert4Rec | $4.96_{\pm 0.05}$ | $2.05_{\pm 0.03}$ | $3.77_{\pm 0.09}$ | $1.73_{\pm 0.04}$ | $2.01_{\pm 0.06}$ | $0.72_{\pm 0.04}$ |
| CL4SRec | $4.67_{\pm 0.03}$ | $2.12_{\pm 0.01}$ | $4.43_{\pm 0.07}$ | $2.66_{\pm 0.03}$ | $2.11_{\pm 0.05}$ | $0.76_{\pm 0.04}$ |
| DiffRec | $4.33_{\pm 0.02}$ | $1.84_{\pm 0.01}$ | $3.74_{\pm 0.08}$ | $1.77_{\pm 0.05}$ | $1.82_{\pm 0.03}$ | $0.65_{\pm 0.09}$ |
| DiffuRec | $4.63_{\pm 0.03}$ | $2.23_{\pm 0.04}$ | $4.51_{\pm 0.02}$ | $3.40_{\pm 0.06}$ | $2.23_{\pm 0.09}$ | $0.81_{\pm 0.05}$ |
| DreamRec | $4.78_{\pm 0.06}$ | $2.23_{\pm 0.02}$ | $\underline{5.16}_{\pm 0.05}$ | $\underline{4.11}_{\pm 0.02}$ | $\underline{2.26}_{\pm 0.07}$ | $0.79_{\pm 0.01}$ |
| DiffuASR | $4.48_{\pm 0.03}$ | $1.92_{\pm 0.02}$ | $4.53_{\pm 0.02}$ | $3.30_{\pm 0.03}$ | $2.05_{\pm 0.02}$ | $0.71_{\pm 0.02}$ |
| PDRec | $5.43_{\pm 0.02}$ | $\underline{3.08}_{\pm 0.02}$ | $4.48_{\pm 0.02}$ | $3.68_{\pm 0.03}$ | $2.12_{\pm 0.04}$ | $0.76_{\pm 0.03}$ |
| DimeRec | $5.33_{\pm 0.03}$ | $3.86_{\pm 0.08}$ | $4.17_{\pm 0.04}$ | $3.64_{\pm 0.05}$ | $2.06_{\pm 0.02}$ | $0.78_{\pm 0.06}$ |
| PreferDiff | $\underline{5.74}_{\pm 0.07}$ | $3.07_{\pm 0.08}$ | $4.87_{\pm 0.02}$ | $3.81_{\pm 0.07}$ | $2.22_{\pm 0.03}$ | $\underline{0.83}_{\pm 0.01}$ |
| Ours | $\mathbf{6.15}_{\pm 0.03}$ | $\mathbf{4.06}_{\pm 0.02}$ | $\mathbf{5.40}_{\pm 0.02}$ | $\mathbf{4.42}_{\pm 0.04}$ | $\mathbf{2.43}_{\pm 0.01}$ | $\mathbf{0.91}_{\pm 0.06}$ |
| improv. | 7.14% | 31.82% | 4.65% | 7.54% | 7.52% | 9.64% |

Table 2: Ablation Study for TCR and APA. The best performance is bolded.

| Methods | YooChoose | | KuaiRec | | Zhihu | |
|---|---|---|---|---|---|---|
| | HR@20 | NDCG@20 | HR@20 | NDCG@20 | HR@20 | NDCG@20 |
| DDPM | $5.68_{\pm 0.05}$ | $3.81_{\pm 0.03}$ | $5.23_{\pm 0.02}$ | $4.23_{\pm 0.06}$ | $2.33_{\pm 0.04}$ | $0.86_{\pm 0.03}$ |
| DDIM | $5.34_{\pm 0.04}$ | $3.75_{\pm 0.03}$ | $5.05_{\pm 0.02}$ | $3.91_{\pm 0.05}$ | $2.38_{\pm 0.01}$ | $0.81_{\pm 0.05}$ |
| w/o TCR | $5.72_{\pm 0.02}$ | $3.82_{\pm 0.03}$ | $5.24_{\pm 0.02}$ | $4.23_{\pm 0.04}$ | $2.39_{\pm 0.05}$ | $0.87_{\pm 0.01}$ |
| w/o APA | $5.89_{\pm 0.02}$ | $3.82_{\pm 0.05}$ | $5.33_{\pm 0.06}$ | $4.30_{\pm 0.02}$ | $2.41_{\pm 0.04}$ | $0.89_{\pm 0.05}$ |
| w/o td | $6.10_{\pm 0.02}$ | $4.04_{\pm 0.04}$ | $5.36_{\pm 0.04}$ | $4.39_{\pm 0.05}$ | $2.28_{\pm 0.05}$ | $0.87_{\pm 0.02}$ |
| w/o d | $6.11_{\pm 0.01}$ | $4.05_{\pm 0.04}$ | $5.38_{\pm 0.03}$ | $4.40_{\pm 0.03}$ | $2.38_{\pm 0.01}$ | $0.89_{\pm 0.02}$ |
| w/o t | $6.12_{\pm 0.03}$ | $4.04_{\pm 0.01}$ | $5.39_{\pm 0.02}$ | $4.41_{\pm 0.03}$ | $2.35_{\pm 0.01}$ | $0.89_{\pm 0.04}$ |
| Ours | $\mathbf{6.15}_{\pm 0.02}$ | $\mathbf{4.06}_{\pm 0.04}$ | $\mathbf{5.40}_{\pm 0.02}$ | $\mathbf{4.42}_{\pm 0.01}$ | $\mathbf{2.43}_{\pm 0.04}$ | $\mathbf{0.91}_{\pm 0.02}$ |

considering step $t$ or pair similarity $s$ or both. As shown in Table 2, all the variants surpass the DDPM- or DDIM-based recommenders, demonstrating the significance of TCR and APA. Additionally, the superiority of "w/o d", "w/o t" over "w/o td" validates the necessity of pair- and step-wise adaptation for APA. More analysis for our ablation study is detailed in Appendix D.1.

## 5.4 Sensitivity Analysis (RQ3)

We investigate the sensitivity of TA-Rec to the hyperparameters $\lambda_c$ (controls the weight of TCR loss in pertaining stage) and $\lambda_\beta$ (controls the strength of preference alignment in fine-tuning stage). The results of $\lambda_c$ are shown in Figure 3, where "onestep" and "multistep" refer to TA-Rec using one-step generation and its extension in a multi-step setting (5-10). We can observe that the performance peaks when the regularization strength $\lambda_c \in [0.4, 0.8]$, balancing consistency smoothing and item reconstruction. Excessive regularization ($\lambda_c > 0.8$) suppresses oracle item modeling, while weak regularization ($\lambda_c < 0.3$) fails to mitigate discretization errors. The results for $\lambda_\beta$ are presented in Figure 4, where "fix_one", "fix_mul", "Our_sone", and "fix_one" refer to fine-tuning TA-Rec with fixed $\lambda_\beta$ or our adaptive $\lambda_\beta(t, d)$ with one or multiple steps-generation, respectively. The curves of "fix_one" and "fix_mul" fluctuate as $\lambda_\beta$ changes, highlighting the significance of $\lambda_\beta$ in performance optimization. The curves of "Ours_one" and "Ours_mul" lie above those of "fix_one" and "fix_mul", validating the superiority of our design of adaptive coefficient $\lambda_\beta(t, d)$ in aligning user preference.

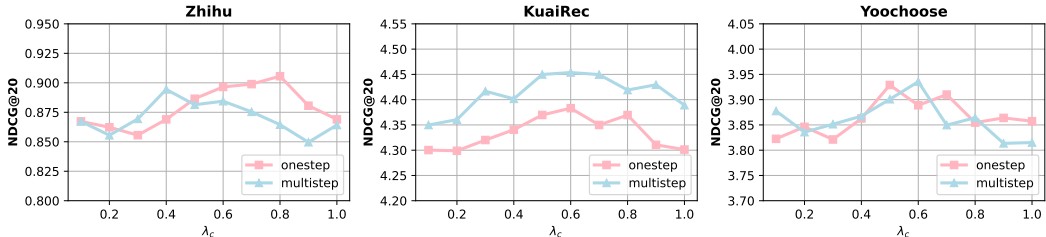

Figure 3: Performance of TA-Rec on different $\lambda_c$, demonstrating the sensitivity of TA-Rec to the strength of consistency regularization.

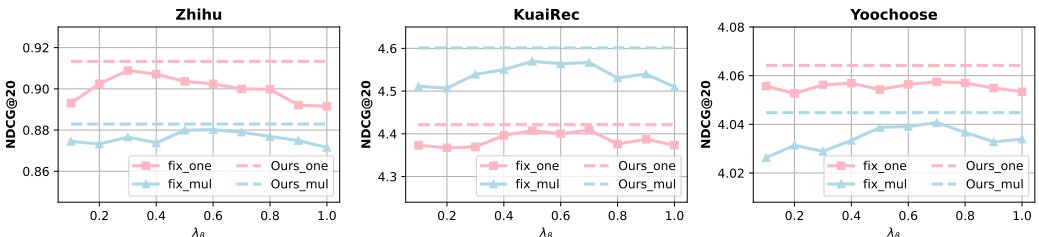

Figure 4: Performance of TA-Rec on different $\lambda_\beta$, demonstrating the sensitivity of TA-Rec on the strength of preference alignment and the superiority of our Adaptive preference alignment.

## 5.5 Computational Resource Comparison (RQ4)

To answer RQ4, we compare the training and inference efficiency of TA-Rec with traditional recommenders (*e.g.,* SASRec [16]) and diffusion-based recommenders (DreamRec [4] with 1k steps and PreferDiff [7] with 10-20 steps). The time costs are presented in Table 3. We observe that the computational complexity of TA-Rec for training each epoch is comparable to that of other diffusion-based and traditional recommenders using the same sequence encoder (*i.e.,* , Transformer [62]). Moreover, by employing TCR to realize one-step generation, we significantly enhance the efficiency of TA-Rec during the inference phase. As a result, TA-Rec substantially reduces inference time compared to DreamRec and PreferDiff, achieving performance levels similar to SASRec.

Table 3: Running time comparison of TA-Rec and other recommenders that use the same sequence encoder on three datasets.

| Methods | YooChoose | | KuaiRec | | Zhihu | |
|---|---|---|---|---|---|---|
| | Traning | Inferencing | Training | Inferencing | Training | Inferencing |
| SASRec | 01m18s | 00m 06s | 02m 07s | 00m 08s | 00m 12s | 00m 01s |
| DreamRec | 01m 19s | 23m 14s | 02m 23s | 28m 02s | 00m 12s | 05m 04s |
| PreferDiff | 01m 18s | 00m 14s | 02m 22s | 00m 32s | 00m 11s | 00m 03s |
| Ours | 01m 18s | 00m 06s | 02m 21s | 00m 07s | 00m 12s | 00m 01s |

The analysis and experiments conducted for RQ5 are detailed in Appendix D.2, demonstrating that TA-Rec generalizes effectively in multi-step settings (1-5 steps) and that APA can enhance the performance of other diffusion-based recommenders using DDPM [1] or DDIM [35] backbones.

## 6 Conclusion

In this paper, we propose a novel two-stage framework, TA-Rec, to address the critical efficiency-effectiveness trade-off in diffusion-based sequential recommenders. To improve efficiency without compromising recommendation performance, TA-Rec integrates Temporal Consistency Regularization (TCR) in the pretraining stage, smoothing the denoising function to realize one-step generation with bounded error. To further enhance effectiveness, TA-Rec fine-tunes the denoising model with Adaptive Preference Alignment (APA) at both pair- and step-wise. Theoretical justification

and extensive experiments validate that TA-Rec can enhance both efficiency and effectiveness of Diffusion-based recommenders, addressing the discretization error-induced trade-off.

# 7 Acknowledgement

This research is supported by the National Natural Science Foundation of China (62572449, 62302321). This research is also supported by the Fundamental Research Funds for the Central Universities (WK2100250065) and the advanced computing resources provided by the Supercomputing Center of the USTC.

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

# A Proofs

## A.1 Temporal Consistency Regularization

**Theorem 1** *(Error Bound for One-Step Generation) We assume that: (i) The smooth denoising function $f_\theta(\mathbf{x}_t, \mathbf{g}, t)$ satisfies Lipschitz continuity with constants $L > 0$: $\|f_\theta(\mathbf{x}_{s_1/\Delta t}, \mathbf{g}, s_1/\Delta t) - f_\theta(\mathbf{x}_{s_2/\Delta t}, \mathbf{g}, s_2/\Delta t)\|_2 \leq L|s_1 - s_2|$ for any time $s_1, s_2 \in [0, 1]$. (ii) For any step $t \in [1, \ldots, T]$ and any time $s \in [t\Delta t, (t+1)\Delta t]$, there exists a constant $C$ such that the following smooth condition holds $\|f_\theta(\mathbf{x}_{s/\Delta t}, \mathbf{g}, s/\Delta t) - f_\theta(\mathbf{x}_t, \mathbf{g}, t)\|_2 \leq C\|f_\theta(\mathbf{x}_{t+1}, \mathbf{g}, t+1) - f_\theta(\mathbf{x}_t, \mathbf{g}, t)\|_2$. Then, since we minimize $L_{pre}$, the error of one-step generation of $\mathbf{x}$ from any $s$ is bounded as follows:*

$$\sup_{s,\mathbf{x}} \|f_\theta(\mathbf{x}_{s/\Delta t}, \mathbf{g}, s/\Delta t) - \mathbf{x}\|_2 = O(\Delta t). \tag{10}$$

**Proof 1** *We estimate the error*

$$\|f_\theta(\mathbf{x}_{s/\Delta t}, \mathbf{g}, s/\Delta t) - \mathbf{x}\|_2 \leq \|f_\theta(\mathbf{x}_{s/\Delta t}, \mathbf{g}, s/\Delta t) - f_\theta(\mathbf{x}_t, \mathbf{g}, t)\|_2 + \|f_\theta(\mathbf{x}_t, \mathbf{g}, t) - \mathbf{x}\|_2$$
$$= \|f_\theta(\mathbf{x}_{s/\Delta t}, \mathbf{g}, s/\Delta t) - f_\theta(\mathbf{x}_t, \mathbf{g}, t)\|_2 + \delta. \tag{11}$$

*The second term $\|f_\theta(\mathbf{x}_t, \mathbf{g}, t) - \mathbf{x}\|_2$ represents denoising model's reconstruction error, bounded by the loss $\mathcal{L}_{diff}$. This error is inherent to denoising model, regardless of whether a multi-step or our accelerated one-step inference is used. Since we minimize $L_{diff}$, we have $\|f_\theta(\mathbf{x}_t, \mathbf{g}, t) - \mathbf{x}\|_2 < \delta$, $\delta$ is arbitrarily small and negligible. Using the Lipschitz condition, we have*

$$\|f_\theta(\mathbf{x}_{s/\Delta t}, \mathbf{g}, s/\Delta t) - f_\theta(\mathbf{x}_t, \mathbf{g}, t)\|_2 \leq C\|f_\theta(\mathbf{x}_{t+1}, \mathbf{g}, t+1) - f_\theta(\mathbf{x}_t, \mathbf{g}, t)\|_2$$
$$\leq LC|(t+1)\Delta t - t\Delta t| \tag{12}$$
$$= O(\Delta t).$$

*Thus, we have $\sup_{s,\mathbf{x}} \|f_\theta(\mathbf{x}_{s/\Delta t}, \mathbf{g}, s/\Delta t) - \mathbf{x}\|_2 = O(\Delta t)$.*

## A.2 Adaptive Preference Alignment

**Theorem 2** *Suppose that the preference of $f_\theta$ and $f_{ref}$ have the following relation: s.t. $f_{ref}(\mathbf{x}_t^+, \mathbf{g}, t)/f_{ref}(\mathbf{x}_t^-, \mathbf{g}, t) \leq f_\theta(\mathbf{x}_t^+, \mathbf{g}, t)/f_\theta(\mathbf{x}_t^-, \mathbf{g}, t) \leq kf_{ref}(\mathbf{x}_t^+, \mathbf{g}, t)/f_{ref}(\mathbf{x}_t^-, \mathbf{g}, t)$, where $k$ is a constant and $k \leq e$. Then with increasing $\lambda_\beta$, the parameter update of $\theta$ from preference optimization becomes more aggressive, i.e., the norm of gradient $\|\nabla_\theta \mathcal{L}_{Diffusion\text{-}DPO}\| \propto \lambda_\beta$.*

**Proof 2** *The gradient of $\mathcal{L}_{Diffusion\text{-}DPO}$ is defined as:*

$$\|\nabla_\theta \mathcal{L}_{Diffusion\text{-}DPO}\| = \frac{1}{1 + e^{\lambda_\beta \Delta r}} \nabla_\theta(\Delta r), \tag{13}$$

*where $\Delta r = \log \frac{f_\theta(\mathbf{x}_t^+, \mathbf{g}, t)}{f_{ref}(\mathbf{x}_t^+, \mathbf{g}, t)} - \log \frac{f_\theta(\mathbf{x}_t^-, \mathbf{g}, t)}{f_{ref}(\mathbf{x}_t^-, \mathbf{g}, t)}$*

*The gradient of $\nabla_\theta \mathcal{L}_{Diffusion\text{-}DPO}$ with respect to $\lambda_\beta$ is:*

$$\nabla_{\lambda_\beta} \nabla_\theta \mathcal{L}_{Diffusion\text{-}DPO} = \frac{1 + e^{\lambda_\beta \Delta r} - \lambda_\beta \Delta r e^{\lambda_\beta \Delta r}}{(1 + e^{\lambda_\beta \Delta r})^2} \nabla_\theta(\Delta r). \tag{14}$$

*Given the initial condition:*

$$\Delta r = \log\left(\frac{f_\theta(\mathbf{x}_t^+, \mathbf{g}, t)/f_\theta(\mathbf{x}_t^-, \mathbf{g}, t)}{f_{ref}(\mathbf{x}_t^+, \mathbf{g}, t)/f_{ref}(\mathbf{x}_t^-, \mathbf{g}, t)}\right) \leq \log k \leq 1. \tag{15}$$

*Thus, we have:*

$$\frac{1 + e^{\lambda_\beta \Delta r} - \lambda_\beta \Delta r e^{\lambda_\beta \Delta r}}{(1 + e^{\lambda_\beta \Delta r})^2} \nabla_\theta(\Delta r) \geq 0. \tag{16}$$

*This implies two cases:*

1. *If $\nabla_\theta \mathcal{L}_{Diffusion\text{-}DPO} \geq 0$, then $\nabla_\theta(\Delta r) \geq 0$ and $\nabla_{\lambda_\beta} \nabla_\theta \mathcal{L}_{Diffusion\text{-}DPO} \geq 0$.*

2. *If $\nabla_\theta \mathcal{L}_{Diffusion\text{-}DPO} \leq 0$, then $\nabla_\theta(\Delta r) \leq 0$ and $\nabla_{\lambda_\beta} \nabla_\theta \mathcal{L}_{Diffusion\text{-}DPO} \leq 0$.*

*That means the norm of gradient $\|\nabla_\theta \mathcal{L}_{Diffusion\text{-}DPO}\| \propto \lambda_\beta$. The parameters of $\theta$ update slowly when the coefficient $\lambda_\beta$ is small.*

## B  Algorithm

Here we list the algorithm of TA-Rec's pretraining, fine-tuning, and inference phase in Algorithm 1, Algorithm 2, and Algorithm 3.

---

**Algorithm 1:** Pretraining stage of TA-Rec

---

**Input:** Interaction sequence $\mathbf{x}^{1:N-1}$, next item $\mathbf{x}$, hyperparameters $\lambda_c$, initialized denoising model $f_\theta(\cdot)$ and Transformer encoder.
**Output:** optimized denoising model $f_\theta(\cdot)$.
1: $\mathbf{g} = \text{Transformer}(\mathbf{x}^{1:N-1})$ Encode the guidance.
2: $t \sim [1, \ldots, T]$                                 ▷ Sample diffusion step.
3: $\mathbf{z} \sim \mathcal{N}(0, \mathbf{I})$                          ▷ Sample Gaussian noise.
4: $\mathbf{x}_t = \sqrt{\bar{\alpha}_t}\mathbf{x} + (1 - \bar{\alpha}_t)\mathbf{z}$              ▷ Add $t$ steps Gaussian noise.
5: $\mathbf{x}_{t-1} = \sqrt{\bar{\alpha}_{t-1}}\mathbf{x} + (1 - \bar{\alpha}_{t-1})\mathbf{z}$     ▷ Add $t-1$ steps Gaussian noise.
6: $\mathcal{L}_{\text{diff}} = \mathbb{E}_{\mathbf{x},\mathbf{g},t}\left[\|f_\theta(\mathbf{x}_t, \mathbf{g}, t) - \mathbf{x}\|_2^2\right]$       ▷ Calculate reconstruction loss.
7: $\mathcal{L}_{\text{TCR}} = \mathbb{E}_{\mathbf{x}_t,\mathbf{g},t}\left[\|f_\theta(\mathbf{x}_t, \mathbf{g}, t) - f_\theta(\mathbf{x}_{t-1}, \mathbf{g}, t-1)\|_2^2\right]$    ▷ Calculate TCR loss.
8: $\mathcal{L}_{\text{pre}} = \mathcal{L}_{\text{diff}} + \lambda_c \cdot \mathcal{L}_{\text{TCR}}$            ▷ Total loss of pertaining stage
9: Update $f_\theta(\cdot)$ and Transformer encoder with $\mathcal{L}_{\text{TCR}}$.

---

**Algorithm 2:** Finetuning stage of TA-Rec

---

**Input:** preference pair $(\mathbf{x}^+, \mathbf{x}^-)$, pretrained denoising model $f_\theta(\cdot)$, guidance $\mathbf{g}$.
**Output:** optimized denoising model $f_\theta(\cdot)$.
1: $t \sim [1, \ldots, T]$                                 ▷ Sample diffusion step.
2: $\mathbf{z} \sim \mathcal{N}(0, \mathbf{I})$                          ▷ Sample Gaussian noise.
3: $\mathbf{x}_t^+ = \sqrt{\bar{\alpha}_t}\mathbf{x}^+ + (1 - \bar{\alpha}_t)\mathbf{z}$      ▷ Add Gaussian noise to positive items.
4: $\mathbf{x}_t^- = \sqrt{\bar{\alpha}_t}\mathbf{x}^- + (1 - \bar{\alpha}_t)\mathbf{z}$      ▷ Add Gaussian noise to negative items.
5: $\lambda_\beta(t, d) = \lambda_{\text{base}} \cdot \left((1 - \frac{t}{T}) + (1 - d(\mathbf{x}^+, \mathbf{x}^-))\right)$    ▷ Calculate adaptive coefficient.
6: $\mathcal{L}_{\text{APA}} \leftarrow$ Equation 9              ▷ Adaptive preference alignment loss.
7: Update $f_\theta(\cdot)$ with $\mathcal{L}_{\text{APA}}$

---

**Algorithm 3:** Inference phase of TA-Rec

---

**Input:** Interaction sequence $\mathbf{x}^{1:N-1}$, optimal denoise model $f_\theta(\cdot)$, Transformer encoder.
**Output:** Oracle item embedding $\mathbf{x}_0$.
1: $\mathbf{x}_T \sim \mathcal{N}(0, \mathbf{I})$                            ▷ Sample Gaussian noise.
2: $\mathbf{g} = \text{Transformer}(\mathbf{x}^{1:N-1})$      ▷ Encode interaction sequence as guidance.
    # One-step generation
3: $\mathbf{x}_0 = f_\theta(\mathbf{x}_T, \mathbf{g}, T)$
    # Multi-step generation
4: **for** $t = T, \ldots, 1$ **do**
5:     $\mathbf{z} \sim \mathcal{N}(\mathbf{0}, \mathbf{I})$ if $t > 1$, else $\mathbf{z} = 0$
6:     $\hat{\mathbf{x}}_0 = f_\theta(\mathbf{x}_t, \mathbf{g}, t)$
7:     $\mathbf{x}_{t-1} = \frac{\sqrt{\bar{\alpha}_{t-1}}\beta_t}{1-\bar{\alpha}_t}\hat{\mathbf{x}}_0 + \frac{\sqrt{\alpha_t}(1-\bar{\alpha}_{t-1})}{1-\bar{\alpha}_t}\mathbf{x}_t + \sqrt{\tilde{\beta}_t}\mathbf{z}.$    ▷ Reverse step by step.
8: **end for**
9: **return** $\mathbf{x}_0$

---

## C  Detailed Experimental Settings

### C.1  Details of Datasets

Here, we present the statistics of our datasets in Table 4.

Table 4: Statistics of the three datasets.

| Dataset | YooChoose | KuaiRec | Zhihu |
|---|---|---|---|
| #sequences | 128,468 | 92,090 | 11,714 |
| #items | 9,514 | 7,261 | 4,838 |
| #interactions | 539,436 | 737,163 | 77,712 |

## C.2 Detailed Baselines

Here, we detailed our baseline methods, including traditional Recommender, diffusion-based Recommender, and preference-based Recommender. **Traditional Recommender:** These methods predict next items by calculating the similarity between candidate items and the interaction sequences, which are modeled using GRU and Transformer architectures.

- GRU4Rec [57] uses gated recurrent units (GRUs) to model user behavior sequences, processing session data sequentially to predict next-item interactions.

- Caser [58] applies horizontal and vertical convolutional filters to capture both point-wise and union-level sequential patterns from user interaction sequences.

- SASRec [16] employs self-attention mechanisms to weight historical interactions dynamically, modeling long-range dependencies in user behavior sequences.

- Bert4Rec [59] adapts the bidirectional Transformer architecture with masked item prediction to learn contextual representations of user behavior sequences.

- CL4SRec [60] incorporates contrastive learning by constructing augmented sequence views through item cropping, masking, and reordering operations.

**Diffusion-based Recommender:** These methods leverage diffusion models to formulate the adding noise and denoising process in generative recommenders, generating item embeddings or item scores step by step.

- DiffRec [6] optimizes the diffusion model to predict noise in corrupted interactions, where diffusion steps progressively refine interaction probabilities.

- DiffuRec [5] employs diffusion models for sequential recommendation, which adds Gaussian noise to next items while preserving historical sequences.

- DreamRec [4] reformulates sequential recommendation as oracle item generation via classifier-free guidance-based diffusion, where encoded historical interactions serve as condition signals.

**Preference-based Recommender:** These methods employ diffusion models to enrich user preference representations or enhance diffusion models' ability with learned user preferences.

- DiffuASR [26] enhances sequential recommendation by generating high-quality pseudo sequences using a diffusion model, specifically designed to better capture and align with user preferences.

- PDRec [17] leverages diffusion models to generate comprehensive user preferences on all items, using strategies like historical behavior reweighting and noise-free negative sampling to enhance the representation of user preferences.

- DimeRec [61] shifts the recommendation task from generating specific items to generating user interests, using a multi-interest model to extract stable user preferences and a diffusion model to reconstruct user embeddings.

- PreferDiff [7] introduces a tailored optimization objective for diffusion-based recommenders, transforming BPR into a log-likelihood ranking objective and integrating multiple negative samples to better capture and align with user preferences

Table 5: Ablation Study for Temporal Consistency Regularization. The best performance is bolded.

| Methods | YooChoose | | KuaiRec | | Zhihu | |
|---------|-----------|-----------|---------|-----------|-------|-----------|
| | HR@20 | NDCG@20 | HR@20 | NDCG@20 | HR@20 | NDCG@20 |
| DDPM | $5.68_{\pm 0.05}$ | $3.81_{\pm 0.03}$ | $5.23_{\pm 0.02}$ | $4.23_{\pm 0.06}$ | $2.33_{\pm 0.04}$ | $0.86_{\pm 0.03}$ |
| DDIM | $5.34_{\pm 0.04}$ | $3.75_{\pm 0.03}$ | $5.05_{\pm 0.02}$ | $3.91_{\pm 0.05}$ | $2.38_{\pm 0.01}$ | $0.81_{\pm 0.05}$ |
| w/o TCR | $5.72_{\pm 0.02}$ | $3.82_{\pm 0.03}$ | $5.24_{\pm 0.02}$ | $4.23_{\pm 0.04}$ | $2.39_{\pm 0.05}$ | $0.87_{\pm 0.01}$ |
| w/ TCR | $5.89_{\pm 0.02}$ | $3.82_{\pm 0.05}$ | $5.33_{\pm 0.06}$ | $4.30_{\pm 0.02}$ | $2.41_{\pm 0.04}$ | $0.89_{\pm 0.05}$ |
| Ours | $\mathbf{6.15}_{\pm 0.02}$ | $\mathbf{4.06}_{\pm 0.04}$ | $\mathbf{5.40}_{\pm 0.02}$ | $\mathbf{4.42}_{\pm 0.01}$ | $\mathbf{2.43}_{\pm 0.04}$ | $\mathbf{0.91}_{\pm 0.02}$ |

Table 6: Ablation Study for the adaptive strategies for preference Alignment.

| Methods | YooChoose | | KuaiRec | | Zhihu | |
|---------|-----------|-----------|---------|-----------|-------|-----------|
| | HR@20 | NDCG@20 | HR@20 | NDCG@20 | HR@20 | NDCG@20 |
| w/o APA | $5.89_{\pm 0.02}$ | $3.82_{\pm 0.05}$ | $5.33_{\pm 0.06}$ | $4.30_{\pm 0.02}$ | $2.41_{\pm 0.04}$ | $0.89_{\pm 0.05}$ |
| w/o td | $6.10_{\pm 0.02}$ | $4.04_{\pm 0.04}$ | $5.36_{\pm 0.04}$ | $4.39_{\pm 0.05}$ | $2.28_{\pm 0.05}$ | $0.87_{\pm 0.02}$ |
| w/o d | $6.11_{\pm 0.01}$ | $4.05_{\pm 0.04}$ | $5.38_{\pm 0.03}$ | $4.40_{\pm 0.03}$ | $2.38_{\pm 0.01}$ | $0.89_{\pm 0.02}$ |
| w/o t | $6.12_{\pm 0.03}$ | $4.04_{\pm 0.01}$ | $5.39_{\pm 0.02}$ | $4.41_{\pm 0.03}$ | $2.35_{\pm 0.01}$ | $0.89_{\pm 0.04}$ |
| Ours | $\mathbf{6.15}_{\pm 0.02}$ | $\mathbf{4.06}_{\pm 0.04}$ | $\mathbf{5.40}_{\pm 0.02}$ | $\mathbf{4.42}_{\pm 0.01}$ | $\mathbf{2.43}_{\pm 0.04}$ | $\mathbf{0.91}_{\pm 0.02}$ |

# D  More Experimental Results

## D.1  Detailed Analysis for Ablation Study

We present the ablation study for TCR and APA in Tables 5 and 6. From the results in Table 5, we observe that "DDPM" achieves moderate performance but incurs high inference costs with a 1,000-step reverse process. In contrast, "DDIM" reduces reverse steps to 10-20 but experiences a performance drop (*e.g.,* $3.75\%$ for "DDIM" vs. $3.81\%$ for "DDPM" on YooChoose). However, "w/ TCR", which employs TCR to facilitate one-step generation through a smoothed denoising function, outperforms both "DDPM" and "DDIM", underscoring TCR's advantage in accelerating generation without sacrificing recommendation performance. Meanwhile, "w/o TCR", which fine-tunes DDPM-based recommendations pretrained without TCR loss, surpasses the "DDPM" variant but lags behind our TA-Rec, further highlighting the importance of TCR loss during the pretraining stage.

Additionally, as shown in Table 6, "w/o td", "w/o d", "w/o t", and our TA-Rec all outperform "w/o APA" across the three datasets, confirming the effectiveness of APA in aligning user preferences during the fine-tuning stage. Removing step-wise adaptation ("w/o t") or pair-wise adaptation ("w/o d") degrades the performance of our TA-Rec (*e.g.,* Zhihu HR@20 drops from $2.43\%$ to $2.35\%$ and $2.38\%$, respectively), demonstrating the necessity of both adaptations for effective alignment. This further shows that dynamically adjusting the alignment strength $\lambda_\beta(t, d)$ based on both timestep $t$ and preference pair similarity $d$ can help mitigate overfitting to ambiguous preferences (*i.e.,* preference pairs with higher similarity $d$) and noise (noisy inputs with larger $t$).

## D.2  Generalization Ability of TA-Rec

To answer RQ5, we generalize TCR to multi-step reverse process settings. The results are presented in Figure 5, where the variants "DDPM" and "DDIM" refer to the diffusion-based sequential recommenders based on DDPM and DDIM backbones, respectively, while "Ours" denotes that based on TCR training. "Our" consistently outperforms both variants across generation settings of 1-5 steps, as indicated by the pink column's superiority over the green and blue columns at various reverse-step settings. Furthermore, the performance of "Ours" remains stable across different reverse-step configurations, validating that TCR generalizes effectively in multi-step generation due to its smooth denoising function.

Additionally, we apply APA to various pre-trained diffusion-based recommenders, with results detailed in Table 7. This includes DDPM-based, DDIM-based, and TCR-based recommenders. For TCR-based recommenders, we evaluate outcomes from one-step and multi-step (5-step) generation.

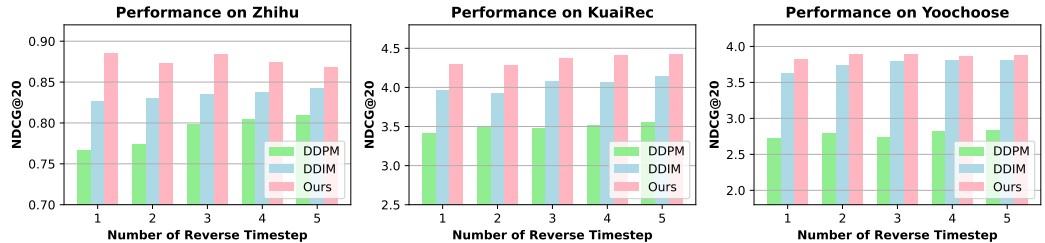

Figure 5: Performance of TA-Rec on multi-step inference settings, demonstrating the generalization ability and robustness of TCR in the multi-step reverse process.

Table 7: Performance of TA-Rec on different pretrained diffusion-based recommenders, demonstrating the generalization ability and effectiveness of APA in enhancing diffusion-based recommenders' effectiveness.

| Methods | YooChoose | | KuaiRec | | Zhihu | |
|---------|-----------|-----------|---------|-----------|-------|-----------|
| | HR@20 | NDCG@20 | HR@20 | NDCG@20 | HR@20 | NDCG@20 |
| DDPM | $5.68_{\pm 0.05}$ | $3.81_{\pm 0.03}$ | $5.23_{\pm 0.02}$ | $4.23_{\pm 0.06}$ | $2.33_{\pm 0.04}$ | $0.86_{\pm 0.03}$ |
| +APA | $5.72_{\pm 0.03}$ | $3.82_{\pm 0.01}$ | $5.24_{\pm 0.03}$ | $4.24_{\pm 0.05}$ | $2.39_{\pm 0.02}$ | $0.87_{\pm 0.04}$ |
| DDIM | $5.34_{\pm 0.04}$ | $3.75_{\pm 0.03}$ | $5.05_{\pm 0.02}$ | $3.91_{\pm 0.05}$ | $2.38_{\pm 0.01}$ | $0.81_{\pm 0.05}$ |
| +APA | $5.37_{\pm 0.03}$ | $3.77_{\pm 0.06}$ | $5.14_{\pm 0.04}$ | $4.07_{\pm 0.05}$ | $2.40_{\pm 0.03}$ | $0.83_{\pm 0.03}$ |
| TCR-1 | $5.89_{\pm 0.02}$ | $3.82_{\pm 0.05}$ | $5.33_{\pm 0.06}$ | $4.30_{\pm 0.02}$ | $2.41_{\pm 0.04}$ | $0.89_{\pm 0.05}$ |
| +APA | $6.15_{\pm 0.02}$ | $4.06_{\pm 0.04}$ | $5.40_{\pm 0.02}$ | $4.42_{\pm 0.01}$ | $2.43_{\pm 0.04}$ | $0.91_{\pm 0.02}$ |
| TCR-m | $5.93_{\pm 0.04}$ | $3.88_{\pm 0.02}$ | $5.38_{\pm 0.03}$ | $4.43_{\pm 0.04}$ | $2.37_{\pm 0.05}$ | $0.86_{\pm 0.05}$ |
| +APA | $6.00_{\pm 0.03}$ | $4.04_{\pm 0.03}$ | $5.40_{\pm 0.03}$ | $4.60_{\pm 0.04}$ | $2.41_{\pm 0.02}$ | $0.88_{\pm 0.04}$ |

As shown in Table 7, applying the APA strategy to these pre-trained diffusion-based recommenders yields improved recommendation performance across all three datasets. For instance, "DDIM+APA" enhances the HR@20 performance of "DDIM" from $5.05\%$ to $5.14\%$ on KuaiRec. This illustrates the generalization capability of APA, which adaptively aligns denoising models with user preferences to enhance the effectiveness of diffusion-based recommenders.

### D.3 Exploration of Different Negative Sampling Strategies of TA-Rec

As presented in Sec 4.4, we randomly sample one item from the item corpus that the user has not interacted with as the negative to construct a preference pair for each user. Since our main contribution is the adaptive alignment strength, which can mitigate overfitting to ambiguous preferences and noise, the random sampling strategy is employed for simplicity.

To explore the impact of different negative sampling strategies, we perform additional experiments on TA-Rec with results presented in Table 8. The "ours-hard" represents selecting hard negatives based on high cosine similarity to the positive item. The "ours-popular" denotes selecting negative items according to item popularity. The slightly superior performance of "ours-popular" and "ours-hard" over "ours" suggests that adopting different negative sampling strategies can be further considered.

### D.4 Analysis on the Diversity of Diffusion-based Recommenders

For diffusion-based recommenders, the mechanism of adding random noise to target items and generating "oracle items" from stochastic noise inherently enhances the diversity of recommendation results. Our model, TA-Rec, preserves this fundamental mechanism and accelerates generation by smoothing the denoising function. The TCR loss is designed to ensure the discretization error remains bounded during this accelerated generation. It does not eliminate the model's dependence on the random noise, so the potential for diverse generation is maintained in TA-Rec.

Table 8: Experiments on different negative sampling strategies.

| Methods | YooChoose | | KuaiRec | | Zhihu | |
|---|---|---|---|---|---|---|
| | HR@20 | NDCG@20 | HR@20 | NDCG@20 | HR@20 | NDCG@20 |
| w/o APA | 5.89 | 3.82 | 5.33 | 4.30 | 2.41 | 0.89 |
| ours | 6.15 | 4.06 | 5.40 | 4.42 | 2.43 | 0.91 |
| ours-popular | 6.17 | 4.07 | 5.43 | 4.43 | 2.43 | 0.92 |
| ours-hard | 6.16 | 4.08 | 5.42 | 4.40 | 2.45 | 0.93 |

Table 9: Experiments on the diversity (coverage@20) of diffusion recommenders.

| Methods | YooChoose | KuaiRec | Zhihu |
|---|---|---|---|
| SASRec | 0.1703 | 0.8368 | 0.7306 |
| DreamRec | 0.2051 | 0.8426 | 0.7616 |
| ours | 0.2042 | 0.8416 | 0.7609 |

To validate the diversity of diffusion-based recommender and our TA-Rec, we conduct additional experiments on the coverage metric, which measures the proportion of unique items recommended across multiple users. The experimental results presented in Table 9 indicate that both TA-Rec and DreamRec, which leverage diffusion models for generating recommendations, exhibit greater diversity compared to traditional recommenders like SASRec. This underscores the advantages of diffusion models in achieving diverse recommendations. Furthermore, the recommendations from TA-Rec show comparable diversity to those from DreamRec, suggesting that the inherent diversity benefits of diffusion models are maintained even with the faster inference speed.

### D.5 Experiments on More Datasets.

Here, we conduct experiments to validate TA-rec on more diverse datasets (Steam [63], Beauty [64], and Toys), varying in sizes and domains. The statistics of these datasets are shown in Table 10. Experimental results are presented in Table R2.

Our method consistently outperforms various baselines on larger dataset (Steam) and diverse datasets (Amazon-beauty and Amazon-toys), further highlighting the effectiveness of TA-Rec.

## E   Discussion and Limitation

### E.1   Relationship with Consistency models

Our Temporal Consistency Regularization (TCR) shares conceptual similarities with consistency models [15] in accelerating diffusion-based generation by enforcing self-consistency across timesteps. However, TCR differs fundamentally in its design and applicability to sequential recommendation scenarios:

**Stochastic vs. Deterministic Trajectories:** Consistency models typically enforce consistency along deterministic ODE trajectories, assuming a smooth and predefined path for generation. In contrast, TCR operates on the stochastic SDE process [2], which inherently models uncertainty in user behavior sequences. This is critical for recommendation systems, where user preferences are noisy and non-deterministic. By regularizing the denoising results of adjacent steps, TCR preserves the stochastic nature of denoising while smoothing the trajectory, making it more robust to the dynamic and noisy patterns of real-world user interactions.

**With vs. without distillation:** Consistency models often require distilling knowledge from a pre-trained diffusion model, assuming the original model has already captured the data distribution. However, sequential recommendation tasks lack unified pretrained backbone models, making the distillation unreliable. In contrast, TCR is jointly optimized with the item reconstruction loss during pretraining, directly aligning the denoising results with oracle items. This end-to-end approach is independent of numerical solvers, eliminating the need for distillation with complex solvers and ensuring the denoising accuracy under consistency regularization.

Table 10: Statistics of the Steam, Beauty, and Toys datasets.

| Dataset | Steam | Beauty | Toys |
|---|---|---|---|
| #sequences | 281,428 | 22,363 | 19,412 |
| #items | 13,044 | 12,101 | 11,924 |
| #interactions | 3,485,022 | 198,502 | 167,597 |

Table 11: The performance of TA-Rec and baseline methods on more datasets(Steam, Beauty, and Toys).

| Methods | Steam | | Toys | | Beauty | |
|---|---|---|---|---|---|---|
| | HR@20 | NDCG@20 | HR@20 | NDCG@20 | HR@20 | NDCG@20 |
| GRU4Rec | 10.13 | 4.21 | 5.54 | 3.45 | 6.46 | 3.48 |
| Caser | 15.12 | 6.42 | 8.53 | 4.21 | 8.35 | 4.21 |
| SASRec | 13.61 | 5.36 | 9.23 | 4.33 | 8.98 | 3.66 |
| Bert4Rec | 12.73 | 5.20 | 7.49 | 4.02 | 8.59 | 3.45 |
| CL4SRec | 15.06 | 6.12 | 9.09 | 5.08 | 10.18 | 4.85 |
| DiffRec | 15.09 | 6.89 | 9.18 | 5.25 | 10.21 | 5.14 |
| DiffuRec | 15.83 | 7.08 | 10.06 | 5.18 | 10.36 | 5.21 |
| DreamRec | 15.08 | 6.39 | 9.88 | 5.22 | 10.32 | 4.88 |
| DiffuASR | 15.74 | 6.59 | 9.39 | 5.19 | 10.03 | 5.16 |
| PDRec | 15.78 | 6.51 | 9.08 | 5.12 | 10.24 | 5.02 |
| DimeRec | 15.29 | 6.45 | 9.15 | 5.24 | 10.46 | 5.44 |
| PreferDiff | 15.92 | 7.12 | 10.18 | 5.34 | 10.69 | 5.38 |
| Ours | 16.25 | 7.36 | 10.87 | 5.81 | 10.99 | 5.54 |
| improv. | 2.07% | 3.37% | 6.78% | 8.80% | 2.81% | 1.84% |

## E.2 Broader Impact

TA-Rec's one-step generation framework substantially reduces the computational overhead of deploying diffusion-based recommender systems in industrial settings. By replacing multi-step denoising with a single-step process while maintaining accuracy, it enables real-time personalization for latency-critical applications such as live-streaming commerce and instant content delivery platforms. These efficiency improvements not only enhance recommendation quality but also boost user engagement and satisfaction in real-world applications.

TA-Rec's approach to accelerating generation while preserving precision provides methodological insights for adapting diffusion models to multimodal recommendation tasks (*e.g.,* text, image, and video hybrids), thereby streamlining their adoption in next-generation AI services. This advancement bridges the gap between theoretical generative models and scalable, industry-ready solutions, particularly in scenarios requiring seamless integration of heterogeneous data types.

## E.3 Limitations and Future Work

**Training Cost Limitation**: While TA-Rec achieves significant inference efficiency through one-step generation, its two-stage training framework (pretraining with TCR and fine-tuning with APA) requires more computational resources compared to end-to-end approaches. Additionally, the TCR necessitates executing the denoising model twice for the adjacent steps per training iteration, thereby incurring additional computational overhead during the pretraining stage.

**Future Work**: We will explore integrating large language model (LLM) scaling laws with diffusion-based generative recommendation frameworks to develop a diffusion LLM for generative recommendation. Specifically, we aim to investigate how LLM emergent capabilities—such as context understanding and semantic reasoning—scale with model size, data volume, and computational resources to enhance the expressive power of diffusion processes in capturing complex user-item interaction patterns.

