# OpenReview forum: "On Efficiency-Effectiveness Trade-off of Diffusion-based Recommenders"
_NeurIPS.cc/2025/Conference — NeurIPS 2025 poster_

### Official Review · Reviewer_ptjv · 2025-06-28

**Clarity:** 1
**Significance:** 3
**Originality:** 1
**Rating:** 4
**Confidence:** 5

**Summary:**

This paper introduces TA-Rec, a novel two-stage framework addressing the critical efficiency-effectiveness trade-off in diffusion-based sequential recommenders. The pretraining stage involves Temporal Consistency Regularization (TCR) to enforce consistency among denoising results, enabling efficient one-step generation of the next item (oracle item). Subsequently, the model undergoes fine-tuning with Adaptive Preference Alignment (APA) to align the denoising process more effectively with user preferences. The authors demonstrate the superiority of their proposed framework in both computational efficiency (inference time) and recommendation effectiveness (performance).

**Questions:**

1. In Table 1, the number of reverse steps is stated only for DreamRec and PreferDiff. Could you clarify the number of reverse steps used by other methods (e.g., DiffRec, DiffuRec) included in the comparison?

2. In Table 3, does the reported training time for TA-Rec encompass the cumulative duration of both the pretraining stage and fine-tuning stage?

3. The paper mentions that the Transformer model is frozen during the fine-tuning stage. Could you explain this design choice? Is there any experimental analysis? Were there any specific reasons that lead to this decision? (e.g., stability, cost)

4. The use of a one-step reverse process, even with guidance embedding, might inherently lead to a decrease in the diversity of generated items or a tendency towards uniformized item embeddings. Could you suggest or conduct additional analyses to asses the diversity of the recommended items or the characteristics of the generated embedding space or item embedding space? (e.g., embedding space visualization techniques like t-SNE)

5. There are some typos and errors.
a. In Table 1, in the case of dataset "YooChoose", the second-best baseline is DimeRec, not PDRec.
b. In Table 3, in the case of dataset "YooChoose", there is typo "traning", not "training".

**Ethical Concerns:**

["NO or VERY MINOR ethics concerns only"]

**Final Justification:**

I thank the authors for their detailed rebuttal and the insightful discussion. While i acknowledge the paper's strengths and strong empirical results, my assessment regarding its limited technical novelty has not changed. In my view, the work represents a high-quality engineering effort but lacks a sufficient conceptual advance to warrant a higher rating. Consequently, I will be keeping my score unchanged.

**Limitations:**

Yes

**Quality:**

3

**Strengths And Weaknesses:**

## Strengths

1. The paper is well-written and well-structured, enhancing its readability and clarity.

2. This work represents a pioneering effort by adopting one-step diffusion within a sequential recommender context. The authors validated the one-step diffusion approach through extensive experiments, demonstrating significant improvements in both speed and accuracy. This work can effectively resolve a core challenge (inference time) in diffusion-based recommenders.

3. The paper proposes a recommender-specific Temporal Consistency Regularization (TCR) to smooth the denoising steps. This crucial component directly enables the feasibility and effectiveness of the one-step diffusion process.

4. The paper provides a mathematical proof for the effectiveness of Adaptive Preference Alignment (APA), an approach developed upon the concept of Direct Preference Optimization (DPO).

## Weaknesses

1. The novelty of this paper is questionable. Given that the Temporal Consistency Regularization (TCR) is derived from the concept of "Consistency Model" and Adaptive Preference Alignment (APA) derived from "Direct Preference Optimization (DPO)", the proposed methodology could be perceived as a refinement of existing concepts rather than a fundamentally new contribution.

2. It would be beneficial to include experiments analyzing the impact of the scale of $\lambda_{base}$. Specifically, demonstrating how performance changes with varying $\lambda_{base}$ value without the time step and similarity terms would provide deeper insights.

3. The premise of Theorem 1, which assumes the pretraining loss as zero, is highly unrealistic. This assumption fundamentally undermines the reliability and practical applicability of the theorem's conclusion, as such an ideal state is unattainable, especially in a real-world recommender scenario.

---

> ### Author Rebuttal · Authors · 2025-07-30
>
> Thanks for your constructive suggestions. We have detailed our responses **point-by-point** to address your concerns.
>
>
> ### Weakness 1:
> > The novelty of this paper is questionable.
>
> Thanks for your question. However, we respectfully argue that our primary contribution is not the invention of these base concepts, but rather the **novel task, technologies designed for recommenders, and theoretical analysis**.
>
> **For tasks**:
> We are the first to identify and address the specific efficiency-effectiveness trade-off in diffusion-based recommenders. This problem arises from the discretization errors inherent in the multi-step generation process.
>
> **For technologies**: Our methods are not direct applications of existing methods but are novelly adapted for the recommendation context.
> - **Novelty in TCR for Recommendation**: Unlike standard Consistency Models which operate on deterministic ODEs, our TCR is designed for the stochastic SDE/DDPM framework, which is crucial for the stochastic nature of user behavior.
> Furthermore, TCR loss is trained jointly with the diffusion reconstruction loss $\mathcal{L}_{diff}$, circumventing the need for a trained teacher diffusion model, which is often impractical in the recommendation domain.
>
> - **Novelty in APA for Diffusion Alignment**: Similarly, our APA is more than a simple application of DPO in diffusion models. Its novelty lies in the introduction of the **adaptive coefficient**, $\lambda_\beta(t,d)$, which dynamically modulates the optimization strength based on both the timestep (noise level) and preference pair similarity. This can avoid diffusion recommenders' overfitting to ambiguous preferences or high noise levels.
>
> **For theoretical analysis**: Our theoretical analysis provides a rigorous foundation for our framework. This includes the analysis of the bounded error of our one-step generation (Theorem 1) and the impact of our adaptive alignment coefficient (Theorem 2), distinguishing our work from a simple application of existing concepts.
>
>
> ### Weakness 2:
> > Analyzing the impact of the scale of $\lambda_{base}$ without the time step and similarity terms.
>
> Thanks for your comment. We have analyzed the sensitivity of TA-Rec to the $\lambda_\beta$ ($\lambda_{base}$) without the time step and similarity terms in Figure 4 of our paper. The methods of ``fix_one`` and ``fix_mul`` denote fine-tuning TA-Rec with
> fixed $\lambda_\beta$ without adapting to timestep or similarity in one or multiple steps-generation. The curves of ``fix-one`` and ``fix-mul`` fluctuate as $\lambda_\beta$ ($\lambda_{base}$) changes, highlighting the significance of $\lambda_{\beta}$ in performance optimization. The curves of ``Ours_one`` and ``Ours_mul``, which consider timestep and similarity terms, lie above those of ``fix_one`` and ``fix_mul``, validating the superiority of our design of adaptive coefficient $\lambda_\beta(t,d)$ in aligning user preference.
>
> ### Weakness 3:
> > The premise of Theorem 1, which assumes the pretraining loss as zero, is highly unrealistic.
>
> Thanks for your comment. We agree that the premise of zero pretraining loss in the original Theorem 1 is an idealization for the sake of clarity, and is unattainable in practice. To address this and enhance the theorem's practical applicability, we have reformulated the error bound to depend on the non-zero pretraining loss:
>
> $$\begin{aligned}
>        ||f_\theta(\mathbf{x}\_{s/\Delta t}, \mathbf{g}, s/\Delta t) - \mathbf{x}||\_2 &\le ||f\_\theta(\mathbf{x}\_{s/\Delta t}, \mathbf{g}, s/\Delta t) - f\_\theta(\mathbf{x}\_{t},\mathbf{g},{t})||\_2 + ||f\_\theta(\mathbf{x}\_{t},\mathbf{g},{t}) - \mathbf{x}||\_2\\\\
>       & \leq ||f\_\theta(\mathbf{x}\_{s/\Delta t}, \mathbf{g}, s/\Delta t) - f\_\theta(\mathbf{x}\_{t},\mathbf{g},{t})||\_2+\delta.
>     \end{aligned}
> $$
>
> The second term $||f\_\theta(\mathbf{x}\_{t},\mathbf{g},{t}) - \mathbf{x}||\_2$ represents denoising model's reconstruction error. This error is inherent to denoising model, regardless of whether a multi-step or our accelerated one-step inference is used. Instead of assuming it is zero, we now bound it by the pretraining loss $\mathcal{L}\_{pre}$. We can have $||f\_\theta(\mathbf{x}\_{t}, \mathbf{g}, t) - \mathbf{x}||\_2<\delta$. Since we minimize the $L\_{pre}$ loss, the error of one-step generation is still bounded.
>
> ### Questions:
> > 1. The number of reverse steps used by other methods (e.g., DiffRec, DiffuRec) in Table 1?
>
> The reverse steps for DiffuRec and DiffRec are chosen in [10,100,500,1000]. They are configured as [100,500,200] for DiffuRec and [100,500,100] for DiffRec, respectively, in the Yoochoose, KuaiRec, and Zhihu datasets.
>
>
> > 2. The reported training time for TA-Rec
>
> The reported training times in Table 3 reflect ​the time cost​​ for each epoch in the independent phase (pretraining/fine-tuning), not the cumulative time for the two stages.
> The per-epoch training time for each stage is comparable to baselines (e.g., SASRec), as the main computational cost in these methods comes from the Transformer encoder.
> The extra computation required in the pre-training stage is negligible, as it only involves a second pass through the lightweight MLP denoising model.
>
>
> > 3. Explanation of the design of the frozen Transformer model during the fine-tuning stage.
>
> **For stability**: Freezing the guidance encoder provides a stable target for the denoising model to align with during preference optimization, reducing fluctuations in performance.
> **For efficiency**: Fine-tuning only the smaller denoising model (MLP) is significantly faster and less computationally expensive.
>
>
> > 4. Additional analysis to assess the diversity of the recommended items.
>
> We recognize that embedding space visualization techniques like t-SNE are ideal for illustrating the diversity of generated items. However, since we are unable to submit images during the rebuttal, we conducted additional experiments using the coverage@20 metric to analyze the diversity of our recommendations, which measures the proportion of unique items recommended across multiple users.
> The experimental results presented in Table R1 indicate that both TA-Rec and DreamRec, which leverage diffusion models for generating recommendations, exhibit greater diversity compared to traditional recommenders like SASRec. This underscores the **advantages of diffusion models in achieving diverse recommendations through stochastic noise**.
> Furthermore, the recommendations from TA-Rec demonstrate comparable diversity to those from DreamRec, suggesting that **the inherent diversity benefits of diffusion models are preserved even with a one-step reverse process**. This is attributed to the diffusion mechanism of random noise injection at each step, along with the noise-to-item mapping in TA-Rec.
>
> **Table R1:**  Experiments on the diversity of diffusion recommenders.
> | Methods | YooChoose  | KuaiRec |   Zhihu |
> | :---: | :---: | :---: | :---: |
> |  | Cover@20| Cover@20| Cover@20|
> |SASRec|0.1703|0.8368|0.7306|
> |DreamRec|0.2051|0.8426|0.7616|
> |ours|0.2042|0.8416|0.7609|
>
>
> > 5. Typos: the second-best baseline in Table 1 and typo "training" in Table 3
>
> Thanks again for your thorough review and feedback on our paper. We will correct these typos in the paper and carefully examine the context to improve our clarity.
>
> We sincerely hope that our response has adequately **addressed your concerns**. If so, we would greatly appreciate your consideration in **raising the score**. If there are any remaining concerns, **please let us know**, and we will **continue to actively address your comments** and **improve our work**.

---

> > ### Comment · Reviewer_ptjv · 2025-08-05
> >
> > Thank you for your answers.
> >
> > Everything is good, but my primary concern is still the novelty. The core methodology is an adaptation of an existing algorithm to a new domain. This requires a significant effort, but lacks a core novel contribution. Since NeurIPS conference is top-tier conference in the field of machine learning, the lack of novelty can be a critical point. For a publication at NeurIPS, a clearer and more significant novel work is expected. This is the primary reason for my borderline accept score.

---

> > > ### Author Response · Authors · 2025-08-05
> > >
> > > Dear Reviewer,
> > >
> > > Thank you for your recognition of our work and for your feedback. We understand the high standards for novelty at NeurIPS, and we appreciate this opportunity to further clarify the core novelty of our work.
> > > We respectfully posit that our work's novelty is **identifying a new problem** and **making novel improvements to existing concepts** to solve the problem, rather than a simple adaptation.
> > > Our core novelty is demonstrated on three distinct levels:
> > >
> > > 1.  **New Problem Formulation**: Our work is the first to systematically identify and model the critical bottleneck in this emerging area: the inherent **trade-off between efficiency and effectiveness induced by discretization error**. This problem had not been thoroughly explored before, which is a significant and novel contribution itself.
> > >
> > > 2.  **New Method Design**: To solve this problem, we designed a two-stage framework with **TCR** and **APA**, which significantly differ from existing methods, rather than direct applications.
> > >     * **Temporal Consistency Regularization (TCR)** is fundamentally different from consistency models. It is designed for the **stochastic SDE process** that characterizes stochastic user behavior, not the deterministic ODEs for which prior consistency models were developed. Furthermore, TCR is optimized **jointly with the reconstruction loss $\mathcal{L}_{diff}$ from diffusion**, unlike consistency models that typically require a pre-trained teacher diffusion model for knowledge distillation.
> > >     * **Adaptive Preference Alignment (APA)** is more than an application of DPO concept. Its core invention is the **novel adaptive coefficient, $\lambda_{\beta}(t,d)$**, which dynamically modulates the optimization strength based on both **timestep** and **preference pair similarity**. This is a novel mechanism specifically designed to handle the challenges of varying noise levels in diffusion models and ambiguous preferences in recommendation data.
> > >
> > > 3.  **Original Theoretical Contributions**: Our proposed methods are supported by a rigorous theoretical foundation. **Theorem 1** provides the error bound for the one-step generation enabled by TCR, and **Theorem 2** offers a formal justification for the effectiveness of the adaptive coefficient in APA. This theoretical analysis is integral to our contribution and validates the rigor of our approach.
> > >
> > > In summary, our work contributes **a new problem formulation** and a new two-stage framework **containing unique, domain-specific mechanisms different from existing concepts**, all supported by a rigorous theoretical foundation. We believe this complete, theoretically-grounded solution, designed specifically to resolve the efficiency-effectiveness trade-off, has a significant impact on the practical deployment of diffusion models in real-world recommendation systems.
> > >
> > > We sincerely hope our additional clarifications have resolved your concerns about novelty. We are very grateful for your time and for helping us improve our paper with your insightful feedback.

---

### Official Review · Reviewer_dntH · 2025-07-02

**Clarity:** 3
**Significance:** 3
**Originality:** 3
**Rating:** 4
**Confidence:** 2

**Summary:**

This paper proposes a two-stage framework called TA-Rec to address the trade-off between efficiency and effectiveness in diffusion-based sequential recommendation models. The framework includes Temporal Consistency Regularization (TCR) in the pretraining stage to smooth the denoising function and enable one-step generation, while introducing Adaptive Preference Alignment (APA) in the fine-tuning stage to align the denoising process more closely with user preferences. Experimental results show that TA-Rec achieves 100× faster generation compared to leading diffusion-based recommendation methods while improving recommendation performance by 10%. Theoretical analysis demonstrates that TA-Rec can bound the error of accelerated generation, thereby enhancing the reliability of recommendations.

**Questions:**

1. The experimental dataset may lack diversity, more experimental dataset would be prefered.

**Ethical Concerns:**

["NO or VERY MINOR ethics concerns only"]

**Final Justification:**

This paper proposes a two-stage framework called TA-Rec to address the trade-off between efficiency and effectiveness in diffusion-based sequential recommendation models. The framework includes Temporal Consistency Regularization (TCR) in the pretraining stage to smooth the denoising function and enable one-step generation, while introducing Adaptive Preference Alignment (APA) in the fine-tuning stage to align the denoising process more closely with user preferences. Experimental results show that TA-Rec achieves 100× faster generation compared to leading diffusion-based recommendation methods while improving recommendation performance by 10%. Theoretical analysis demonstrates that TA-Rec can bound the error of accelerated generation, thereby enhancing the reliability of recommendations.

APA dynamically adjusts the alignment strength based on preference pair similarity and timesteps during fine-tuning, enhancing recommendation accuracy. This framework effectively addresses the efficiency-effectiveness trade-off in diffusion-based recommendation models. The paper demonstrates that TA-Rec achieves 100× faster generation compared to leading diffusion-based recommenders (e.g., DreamRec) while improving recommendation performance by 10%.  The paper provides theoretical analysis, proving that TA-Rec's one-step generation has bounded error and that its effectiveness aligns with user preferences. This theoretical foundation reinforces the credibility of the framework and its applicability. This paper is well-written and easy to follow. This paper is well-motivated and verified.

**Limitations:**

The TCR module necessitates executing the denoising model twice per training iteration for adjacent steps, increasing computational overhead during the pretraining stage.

**Quality:**

3

**Strengths And Weaknesses:**

Strength:
1. APA dynamically adjusts the alignment strength based on preference pair similarity and timesteps during fine-tuning, enhancing recommendation accuracy. This framework effectively addresses the efficiency-effectiveness trade-off in diffusion-based recommendation models.
2. The paper demonstrates that TA-Rec achieves 100× faster generation compared to leading diffusion-based recommenders (e.g., DreamRec) while improving recommendation performance by 10%.
3. The paper provides theoretical analysis, proving that TA-Rec's one-step generation has bounded error and that its effectiveness aligns with user preferences. This theoretical foundation reinforces the credibility of the framework and its applicability.
4. This paper is well-written and easy to follow.
5. This paper is well-motivated and verified.

Weakness:
1. the TCR module necessitates executing the denoising model twice per training iteration for adjacent steps, increasing computational overhead during the pretraining stage.
2. The experimental dataset may lack diversity, more experimental dataset would be prefered.

---

> ### Author Rebuttal · Authors · 2025-07-30
>
> Thanks for your constructive suggestions. We have detailed our responses **point-by-point** to address your concerns.
>
>
> ### Weakness 1
> > Computational cost of executing the denoising model twice per training iteration for adjacent steps due to TCR loss.
>
> #### Training Complexity
> * **Previous Diffusion Recommender (e.g., DreamRec):** The reconstruction loss $\mathcal{L}_{diff}$ requires guidance encoding from the Transformer and a denoising process through the MLP denoising model. The training complexity is $O (N \cdot L^2 \cdot D + N \cdot D^2)$.
> * **TA-Rec:**
> The ​​TCR loss​​ adds one extra MLP forward pass for an adjacent timestep. The training complexity of TA-Rec is $O(N \cdot L^2 \cdot D +2* N \cdot D^2)$, which is comparable to previous diffusion recommender since the complexity of MLP is negligible vs. Transformer.
>
> **Table 3 in our paper provides a comparative analysis of training time costs, demonstrating the efficiency of TA-Rec**. Importantly, the integration of TCR loss during training facilitates **one-step generation during inference**, thereby significantly enhancing the efficiency of diffusion-based generative recommendation in real-world deployment.
>
>
>
> ### Weakness 2
> > Experiments on more diverse datasets.
>
>  Here, we **conduct experiments** to validate TA-rec on more diverse datasets (Steam, Beauty, and Toys), varying in sizes and domains. The statistics of these datasets are shown in Table R1. Experimental results are presented in **Table R2**.
>
> **Table R1:**  Statistics of the Steam, Beauty, and Toys datasets.
> |             |   Sequence  |     Items    |Ineteractions|
> | ----------- | ----------- | ------------ |------------ |
> |    Steam    |   281,428   |    13,044    | 3,485,022   |
> |   Beauty    |   22,363    |    12,101    |   198,502   |
> |    Toys     |   19,412    |    11,924    |   167,597   |
>
> **Table R2:** Overall performance of different methods for the sequential recommendation on diverse datasets.
> |Methods|Steam||Toys||Beauty||
> |:------------|:------------------:|:------------------:|:------------------:|:------------------:|:------------------:|:------------------:|
> |  | H@20| N@20 | H@20| N@20 | H@20| N@20 |
> |GRU4Rec|10.13|4.21|5.54|3.45|6.46|3.48|
> |Caser|15.12|6.42|8.53|4.21|8.35|4.21|
> |SASRec|13.61|5.36|9.23|4.33|8.98|3.66|
> |Bert4Rec|12.73|5.20|7.49|4.02|8.59|3.45|
> |CL4SRec|15.06|6.12|9.09|5.08|10.18|4.85|
> |DiffRec|15.09|6.89|9.18|5.25|10.21|5.14|
> |DiffuRec|15.83|7.08|10.06|5.18|10.36|5.21|
> |DreamRec|15.08|6.39|9.88|5.22|10.32|4.88|
> |DiffuASR|15.74|6.59|9.39|5.19|10.03|5.16|
> |PDRec|15.78|6.51|9.08|5.12|10.24|5.02|
> |DimeRec|15.29|6.45|9.15|5.24|10.46|5.44|
> |PreferDiff|15.92|7.12|10.18|5.34|10.69|5.38|
> |Ours|16.25|7.36|10.87|5.81|10.99|5.54|
> |improv.|2.07%|3.37%|6.78%|8.80%|2.81%|1.84%|
>
> Our method consistently outperforms various baselines on **larger dataset** (Steam) and **diverse datasets** (Amazon-beauty and Amazon-toys), **further highlighting the effectiveness of TA-Rec**.
>
>
> We sincerely hope that our response has adequately **addressed your concerns**. If so, we would greatly appreciate your consideration in **raising the score**. If there are any remaining concerns, **please let us know**, and we will **continue to actively address your comments** and **improve our work**.

---

> > ### Comment · Reviewer_dntH · 2025-08-08
> >
> > While the rebuttals addressed my main concerns, I believe a Borderline Accept score remains fair for this paper. I will therefore maintain my original score.

---

> > > ### Author Response · Authors · 2025-08-08
> > >
> > > Dear Reviewer,
> > >
> > > We thank you for your valuable time and final feedback. We are pleased that our rebuttal **have addressed your main concerns**, and we particularly appreciate your recognition of our work's strengths, including its motivation, performance improvements, and theoretical foundation.
> > >
> > > We fully respect the reviewer's decision to maintain their score of borderline accept.
> > >
> > > Thank you again.

---

> ### Comment · Area_Chair_hNZE · 2025-08-04
>
> Hi reviewer dntH, This is a gentle reminder that the authors added their response on your question. Can you provide your feedback on their response? Please keep in mind that the deadline of August 6th approaching, and your additional timely feedback greatly enhance further discussions if needed.
>
> Thanks, Area Chair.

---

> ### Author Response · Authors · 2025-08-07
> **Looking forward to your reply**
>
> Dear Reviewer dntH,
>
> Thank you once again for your constructive comments. We have thoroughly addressed your concerns by incorporating new datasets (Steam, Beauty, and Toys) and clarifying the computational costs associated with training during the pretraining stage.
>
> We look forward to further discussion with you and would greatly appreciate your positive feedback on our rebuttal.
>
> Best regards,
>
> The Authors

---

### Official Review · Reviewer_J9pE · 2025-07-03

**Clarity:** 3
**Significance:** 2
**Originality:** 3
**Rating:** 4
**Confidence:** 3

**Summary:**

This paper addresses diffusion-based sequential recommendation systems. Unlike existing models that require many steps (e.g., 1,000 steps) to improve accuracy, this work reduces the process to a single step to achieve efficiency. The model consists of a pretraining phase called the temporal consistency regularizer (TCR) and a finetuning phase called adaptive preference alignment (APA), where the system is further trained to better match user preferences. Compared to prior diffusion models, it is significantly faster while also improving accuracy.

**Questions:**

1. I think there is a possibility that the advantage of diversity inherent in diffusion models has been reduced in exchange for faster inference speed. Why was no analysis or experiment on this aspect included in the paper?
2. As mentioned earlier, if TCR removes the differences between steps to enable 1-step generation, I am curious whether the diffusion process is really functioning as intended. In the ablation study, it is shown that the performance of w/o APA is higher than DDPM and DDIM, which involve multi-step operations. It is puzzling how a single-step generation can outperform traditional multi-step diffusion methods.

**Ethical Concerns:**

["NO or VERY MINOR ethics concerns only"]

**Final Justification:**

My concerns about diversity loss and the necessity of diffusion in 1-step generation were addressed through additional coverage@20 experiments and clear theoretical explanations. The paper is technically solid, with strong experimental validation and a well-structured presentation. TCR and APA are effectively adapted for the recommender system context, and the results demonstrate both efficiency and effectiveness gains.

**Limitations:**

While the authors have thoroughly evaluated the effectiveness and efficiency of their proposed approach, there is no discussion of potential negative societal impacts.

**Paper Formatting Concerns:**

No formatting concerns

**Quality:**

2

**Strengths And Weaknesses:**

- **Quality**: The paper’s contributions are clear, and the theoretical analysis appears to be well-grounded. However, it seems to lack an analysis of the diversity typically associated with diffusion-based models, which could be an area for improvement.
- **Clarity**: The structure and flow of the paper are clear, and the figures and tables are well-organized.
- **Significance**: By substantially improving the efficiency of diffusion-based models—whose main drawback is slow inference—the work makes a meaningful contribution. However, if TCR essentially removes inter-step differences to enable 1-step generation, it raises the question of whether using a diffusion model is necessary at all. The core of diffusion is to generate the desired distribution through denoising noise; if the noise and the output are almost indistinguishable, it is questionable whether diffusion is truly being applied effectively.
- **Originality**: The finetuning process using adaptive preference alignment is well-tailored to recommendation systems, and the resulting performance improvements are notable.

---

> ### Author Rebuttal · Authors · 2025-07-30
>
> Thanks for your constructive suggestions. We have detailed our responses **point-by-point** to address your concerns.
>
>
> ### Weakness 1 and Question 1:
> > Analysis of the diversity associated with diffusion-based models
>
> For diffusion models, the mechanism of **adding random noise** to target items and **generating "oracle items" from stochastic noise** inherently enhances the diversity of recommendation results.
> Our model, TA-Rec, preserves this fundamental mechanism and accelerates generation by smoothing the denoising function. The TCR loss is designed to ensure the discretization error remains bounded during this accelerated generation. It does not eliminate the model's dependence on the random noise,
> so the potential for diverse generation is maintained in TA-Rec.
>
> To validate the diversity of diffusion-based recommender and our TA-Rec, we conduct additional experiments on the coverage metric, which measures the proportion of unique items recommended across multiple users
> The experimental results presented in Table R1 indicate that both TA-Rec and DreamRec, which leverage diffusion models for generating recommendations, exhibit greater diversity compared to traditional recommenders like SASRec. **This underscores the advantages of diffusion models in achieving diverse recommendations**. Furthermore, the recommendations from TA-Rec show comparable diversity to those from DreamRec, suggesting that **the inherent diversity benefits of diffusion models are maintained even with the faster inference speed**.
>
> **Table R1:**  Experiments on the diversity of diffusion recommenders.
> | Methods | YooChoose  | KuaiRec |   Zhihu |
> | :---: | :---: | :---: | :---: |
> |  | Cover@20| Cover@20| Cover@20|
> |SASRec|0.1703|0.8368|0.7306|
> |DreamRec|0.2051|0.8426|0.7616|
> |ours|0.2042|0.8416|0.7609|
>
>
>
> ### Weakness 2 and Question 2:
> > Removing inter-step differences to enable 1-step generation raises the question of whether using a diffusion model is necessary.
>
>
> (1) **Diffusion is a superior training mechanism**: The core mechanism of diffusion models involves **adding random noise to items** and then training a model to **generate "oracle items" from stochastic noise**. We consider this a powerful training mechanism that enriches item distribution, facilitates the learning of item representations, and generates high-quality items to recommend.
> While effective, the multi-step reverse process often leads to significant inference overhead.
> Our proposed model, TA-Rec, accelerates the reverse process to a single denoising step with TCR while **adhering to the fundamental diffusion mechanism**. The value of the diffusion mechanism is empirically demonstrated by comparing TA-Rec to a strong baseline like SASRec (directly predicts next items without diffusion processing). As shown in Table 1 of our paper, TA-Rec significantly outperforms SASRec, even though both utilize similar core modules (e.g., Transformer, MLP). This performance gap **highlights the necessity of the diffusion process for learning higher-quality representations and generating oracle items**.
>
> (2) **Whether the noise and generation are indistinguishable in TA-Rec**: The one-step generation of TA-Rec is achieved by smoothing the denoising function $f_\theta$ via Temporal Consistency Regularization (TCR), which enforces $f_\theta(x_t,t,g)\approx f_\theta(x_{t-1},t-1,g)$ during the noise-to-output mapping. The TCR does not eliminate inter-step differences between $x_t$, $x_{t-1}$, and $x_0$, but strategically smooths the $f_\theta$ to **ensure consistent denoising output $x_0$ across different steps $t$**. This aligns with the concepts of "Consistency Models".
>
>
> (3) **Why our one-step generation outperforms multi-step samplers DDPM/DDIM**: DDPM/DDIM approximate a continuous reverse-SDE process with **finite steps** $T$ (typically 1000). This introduces the **truncation errors that increase as T decreases** (as discussed in Definition 1 of our paper). Even with $T=1000$, DDPM/DDIM converges to a ​​lossy approximation​​ of the true data distribution $P(x_0)$. For TA-Rec, the Temporal Consistency Regularization (TCR) enforces $f_\theta(x_t,t,g) - f_\theta(x_{t-1},t-1,g)\leq \delta$ to smooth the denoising trajectory. By maintaining output consistency across all adjacent steps, TCR ensures that $f_\theta(x_T, T,g)\approx f_\theta(x_t,t,g) \approx x_0 $. Thus, ``w/ APA`` with one-step generation ​​implicitly integrates the infinite-step SDE (infinite $T$)​​, alleviating truncation errors and achieving better performance.
>
>
>
> ### Limitation:
> > Discussion of potential negative societal impacts.
>
> Over-personalization can lead to filter bubbles or echo chambers. Immediate feedback mechanisms may encourage users to engage compulsively, which could negatively impact their mental health.
>
>
>
> We sincerely hope that our response has adequately **addressed your concerns**. If so, we would greatly appreciate your consideration in **raising the score**. If there are any remaining concerns, **please let us know**, and we will **continue to actively address your comments** and **improve our work**.

---

> > ### Comment · Reviewer_J9pE · 2025-08-08
> >
> > Thank you for the rebuttal. My concern regarding 1-step generation has been well addressed through additional experiments and clarification. I will raise my score accordingly.

---

> > > ### Author Response · Authors · 2025-08-08
> > >
> > > Dear reviewer J9pE,
> > >
> > > We sincerely thank you for reconsidering our work and for **your positive feedback**. We are very pleased to hear that our rebuttal and the additional experiments have successfully addressed your concerns regarding 1-step generation.
> > >
> > > Thanks again!

---

> ### Comment · Area_Chair_hNZE · 2025-08-04
>
> Hi reviewer J9pE,
> This is a gentle reminder that the authors added their response on your question.
> Can you provide your feedback on their response?
> Please keep in mind that the deadline of August 6th approaching, and your additional timely feedback greatly enhance further discussions if needed.
>
> Thanks,
> Area Chair.

---

> ### Author Response · Authors · 2025-08-07
> **Looking forward to your reply**
>
> Dear Reviewer J9pE,
>
> Thank you again for your constructive comments. We have thoroughly addressed your concerns by analyzing the diversity of diffusion models and our one-step TA-Rec with additional experiments, and clarifying the advantage and necessity of diffusion training mechanisms.
>
> We look forward to further discussion with you and would greatly appreciate your positive feedback on our rebuttal.
>
> Best regards,
>
> The Authors

---

### Official Review · Reviewer_NQjp · 2025-07-05

**Clarity:** 3
**Significance:** 2
**Originality:** 3
**Rating:** 3
**Confidence:** 2

**Summary:**

This paper introduces a two-stage diffusion framework that achieves one-step generation via Temporal Consistency Regularization (TCR) during pre-training and reduces trajectory deviation through Adaptive Preference Alignment (APA) in fine-tuning. Extensive experiments demonstrate the method’s effectiveness in mitigating discretization-error-induced trade-offs, supported by theoretical analysis proving bounded error for the proposed TA-Rec model.

**Questions:**

1. Could you provide heuristic guidance for selecting important hyper-parameters, such as λ_β_base?
2. What specific negative sampling strategy  is used in APA?
Have alternative strategies been explored? If so, how do they influence performance ?
3. APA shows limited improvement over other diffusion recommenders based on table 7. Are there any further discussion and analysis?
​

​

**Ethical Concerns:**

["NO or VERY MINOR ethics concerns only"]

**Limitations:**

1. ​​Dataset diversity insufficiency​​, limiting claims of broad applicability.
​​2. Omitted computational complexity analysis​​, hindering practical adoption assessment.

**Quality:**

2

**Strengths And Weaknesses:**

Pros:

1. The integration of TCR and APA offers a novel approach to balancing generation efficiency and trajectory alignment, addressing a significant challenge in diffusion-based recommenders.
2. The bounded-error analysis provides a theretical foundation for the method’s rationality.
3. Experiments from multiple angles (e.g., ablation studies, comparison with baselines) convincingly validate the design choices.
4. The paper is well-structured, with clear exposition of motivations, methodology, and results.

Cons:

1. Experiments are conducted on only ​​three datasets​​ of similar scales. To ensure generalizability, validation on more diverse datasets  is essential .
2. The theretical computational costs (time/memory) of TA-Rec, especially during training, are ​​not quantified​​. As scalability is critical for real-world deployment, asymptotic complexity analysis .
3. The negative sampling strategy for APA lacks ​​implementation details​​ and ​​sensitivity analysis​​. Its impact on preference alignment efficacy requires rigorous ablation.

---

> ### Author Rebuttal · Authors · 2025-07-30
>
> Thanks for your constructive suggestions. We have detailed our responses **point-by-point** to address your concerns.
>
> ### Weakness 1
> > Experiments on more diverse datasets.
>
> Here, we **conduct experiments** to validate TA-rec on more diverse datasets (Steam, Beauty, and Toys), varying in sizes and domains. The statistics of these datasets are shown in Table R1. Experimental results are presented in Table R2.
>
> **Table R1:**  Statistics of the Steam, Beauty, and Toys datasets.
> |             |   Sequence  |     Items    |Ineteractions|
> | ----------- | ----------- | ------------ |------------ |
> |    Steam    |   281,428   |    13,044    | 3,485,022   |
> |   Beauty    |   22,363    |    12,101    |   198,502   |
> |    Toys     |   19,412    |    11,924    |   167,597   |
>
> **Table R2:** Overall performance of different methods for the sequential recommendation on diverse datasets.
> |Methods|Steam||Toys||Beauty||
> |:------------|:------------------:|:------------------:|:------------------:|:------------------:|:------------------:|:------------------:|
> |  | H@20| N@20 | H@20| N@20 | H@20| N@20 |
> |GRU4Rec|10.13|4.21|5.54|3.45|6.46|3.48|
> |Caser|15.12|6.42|8.53|4.21|8.35|4.21|
> |SASRec|13.61|5.36|9.23|4.33|8.98|3.66|
> |Bert4Rec|12.73|5.20|7.49|4.02|8.59|3.45|
> |CL4SRec|15.06|6.12|9.09|5.08|10.18|4.85|
> |DiffRec|15.09|6.89|9.18|5.25|10.21|5.14|
> |DiffuRec|15.83|7.08|10.06|5.18|10.36|5.21|
> |DreamRec|15.08|6.39|9.88|5.22|10.32|4.88|
> |DiffuASR|15.74|6.59|9.39|5.19|10.03|5.16|
> |PDRec|15.78|6.51|9.08|5.12|10.24|5.02|
> |DimeRec|15.29|6.45|9.15|5.24|10.46|5.44|
> |PreferDiff|15.92|7.12|10.18|5.34|10.69|5.38|
> |Ours|16.25|7.36|10.87|5.81|10.99|5.54|
> |improv.|2.07%|3.37%|6.78%|8.80%|2.81%|1.84%|
>
> Our method consistently outperforms various baselines on **larger dataset** (Steam) and **diverse datasets** (Amazon-beauty and Amazon-toys), **further highlighting the effectiveness of TA-Rec**.
>
>
> ### Weakness 2
> > Theoretical computational costs of TA-Rec.
>
> Let $N$ be the batch size, $L$ be the sequence length, and $D$ be the embedding dimension. The primary components of our model are the Transformer encoder and the MLP-based denoising model.
>
> **Inference Complexity**
>
> The primary contribution of TA-Rec is the significant reduction in inference complexity compared to prior diffusion models.
>
> * **Traditional Recommender (with Transformer and MLP in model design):**  The complexity of self-attention in Transformer is $O(N \cdot L^2 \cdot D )$, and the complexity of MLP is $O(N \cdot D^2)$, so the total inference complexity is $O(N \cdot L^2 \cdot D + N \cdot D^2)$.
>
>
> * **Previous Diffusion Recommender (e.g., DreamRec):** The complexity is dependent on the number of denoising steps ($T$), which is typically large (~1000).
>     * **Complexity Formula:** $O(T\cdot(N \cdot L^2 \cdot D +  \cdot N \cdot D^2))$
> * **TA-Rec:** By enabling one-step generation ($T=1$), the inference complexity is no longer dependent on a large number of steps and becomes comparable to traditional Transformer-based models.
>     * **Complexity Formula:** $O(N \cdot L^2 \cdot D + N \cdot D^2)$
>
> Thus, the inference complexity of TA-Rec is comparable to traditional transformer-based recommenders (e.g., SASRec).
>
> **Training Complexity:**
> * **Traditional Recommender (e.g., SASRec):** Due to Transformer and MLP modules, the training complexity is $O(N \cdot L^2 \cdot D + N \cdot D^2)$.
>
> * **Previous Diffusion Recommender (e.g., DreamRec):** The reconstruction loss $\mathcal{L}_{diff}$ requires guidance encoding from the Transformer and a denoising process through the MLP denoising model. The training complexity is $O( N \cdot L^2 \cdot D + N \cdot D^2)$.
> * **TA-Rec:**
> The ​​TCR loss​​ adds one extra MLP forward pass for an adjacent timestep. The training complexity of TA-Rec is $O(N \cdot L^2 \cdot D +2* N \cdot D^2)$, which is comparable to traditional recommender since the complexity of MLP is negligible vs. Transformer.
>
> **The time cost comparison in Table 3 of our paper validates the efficiency of TA-Rec for real-world deployment**.
>
>
>
> ### Weakness 3 & Question 2
> > Implementation details and ​sensitivity analysis​ for the negative sampling strategy for APA.
>
> - As presented in Sec 4.4 of our paper, we randomly sample one item from the item corpus that the user has not interacted with as the negative to construct a preference pair for each user. Since our main contribution is the **adaptive alignment strength** which can mitigate overfitting to ambiguous preferences and noise, the random sampling strategy is employed for simplicity.
>
> - To conduct ​sensitivity analysis on the negative sampling strategy, we perform additional experiments to explore different negative sampling strategies in Table R3. The ``ours-hard`` represents selecting hard negatives based on high cosine similarity to the positive item. The ``ours-popular`` denotes selecting negative items according to item popularity. The slightly superior performance of ``ours-popular`` and ``ours-hard`` over ``ours`` suggests that adopting different negative sampling strategies can be further considered.
>
> **Table R3:**  Experiments on hard negative sampling strategy.
> | Methods | YooChoose |  | KuaiRec |  | Zhihu |  |
> | :---: | :---: | :---: | :---: | :---: | :---: | :---: |
> |  | H@20| N@20| H@20| N@20| H@20| N@20|
> |w/o APA| 5.89|3.82| 5.33|4.30|2.41|0.89|
> |ours|6.15|4.06|5.40|4.42|2.43|0.91|
> |ours-popular|6.17|4.07|5.43|4.43|2.43|0.92|
> |ours-hard|6.16|4.08|5.42|4.40|2.45|0.93|
>
>
>
>
> ### Questions
> > Heuristic guidance for selecting $\lambda_{base}$.
>
> - In DPO loss, the hyperparameter $\lambda_\beta$ typically ranges from 0.01 to 0.9. As shown in Figure 3 of our paper, the alignment strength $\lambda_\beta=1/2$ can achieve optimal performance across diverse datasets. As a result, we set $\lambda_{base}=1/2$ and vary $\lambda_\beta(t,d)$ from 0 to 1 with the adaptive mechanism.
>
> > Limited improvement of APA over other diffusion recommenders in Table 7.
> - After further calculations and validations, the performance improvement of APA over other diffusion recommenders ranges from 1% to 4%. All results are the average of 5 independent runs, and these improvements are statistically significant (p-value < 0.05). The improvements observed in TCR and DDIM exceed those in DDPM, indicating that APA may be more effective in enhancing the accuracy of methods designed to accelerate generation.
>
>
> We sincerely hope that our response has adequately **addressed your concerns**. If so, we would greatly appreciate your consideration in **raising the score**. If there are any remaining concerns, **please let us know**, and we will **continue to actively address your comments** and **improve our work**.

---

> > ### Author Response · Authors · 2025-08-08
> >
> > Dear Reviewer NQjp,
> >
> > Thank you again for your constructive comments. As **the discussion period is nearing its end**, we look forward to further discussion with you. If you have any additional questions or concerns, please do not hesitate to reach out. If our rebuttal has adequately addressed your concerns, we would greatly appreciate your positive feedback.
> >
> > Best regards,
> >
> > The Authors

---

> ### Comment · Area_Chair_hNZE · 2025-08-04
>
> Hi reviewer NQjp,
> This is a gentle reminder that the authors added their response on your question.
> Can you provide your feedback on their response?
> Please keep in mind that the deadline of August 6th approaching, and your additional timely feedback greatly enhance further discussions if needed.
>
> Thanks,
> Area Chair.

---

> ### Author Response · Authors · 2025-08-07
> **Looking forward to your reply**
>
> Dear Reviewer NQjp,
>
> Thank you again for your constructive comments. We have thoroughly addressed your concerns by incorporating new datasets (steam, beauty, and toys), providing theoretical computational costs in addition to experimental time costs in our paper, exploring additional negative sampling strategies for APA, and clarifying the selection guidance for $\lambda_base$.
>
> We look forward to further discussion with you and would greatly appreciate your positive feedback on our rebuttal.
>
> Best regards,
>
> The Authors

---

### Official Review · Reviewer_BncC · 2025-07-15

**Clarity:** 3
**Significance:** 3
**Originality:** 3
**Rating:** 4
**Confidence:** 3

**Summary:**

This work develop a two-stage diffusion-based recommendation framework to achieve the trade-off between computational efficiency and recommendation effectiveness, which achieve one-step generation by smoothing the denoising function during pretraining while alleviating trajectory deviation by aligning with user preferences during fine-tuning.

**Questions:**

- If user interaction behavior is sparse, how would this affect the performance of the proposed method? Could it lead to less diverse recommendations?
- Given the stringent inference time constraints in online systems, how can this method be effectively deployed in production environments?
- The authors should further discuss the reasonableness of the Lipschitz continuity assumption when applied to real-world data. Does this assumption hold in practical scenarios, and if not, how might it impact the model's performance?
- Regarding online deployment, does the proposed algorithm support incremental/streaming learning to accommodate real-time recommendation updates based on new user interactions?

**Ethical Concerns:**

["NO or VERY MINOR ethics concerns only"]

**Final Justification:**

I have read the author rebuttal and considered all raised points., I have engaged in discussions and responded to authors. While the rebuttals addressed my main concerns, I believe a Borderline Accept score remains fair for this paper. I will therefore maintain my original score.

**Limitations:**

See Weaknesses and Questions.

**Paper Formatting Concerns:**

There are no major formatting issues in this paper.

**Quality:**

3

**Strengths And Weaknesses:**

Strengths
- The framework with Temporal Consistency Regularization improves the efficiency without sacrificing the recommendation performance by enforcing the consistency between the denoising results across adjacent steps.
- TA-Rec introduces Adaptive Preference Alignment module that aligns the denoising process with user preference adaptively based on preference pair similarity and timesteps.

Weaknesses
- Compared with the traditional Recommenders, the developed two-stage diffusion-based recommendation framework need calculate the dot product between x0 and each candidate item in the corpus, and then recommend K items having the highest similarity scores, which requires more computational resources and may limit its application in small-scale scenarios.
- APA relies on random negative sampling, which may fail to capture hard negative samples, thereby affecting the effectiveness of preference alignment.
- This work does not thoroughly explore the method's performance in data-sparse or cold-start scenarios.

---

> ### Author Rebuttal · Authors · 2025-07-30
>
> Thanks for your constructive suggestions. We have detailed our responses **point-by-point** to address your concerns.
>
> ### Weakness 1:
> > The computational resources for the dot product between $x_0$ and candidate items
>
> - **Calculating similarity between $x_0$ and item corpus then recommending the top-K is a standard final step in recommendation systems (e.g., SASRec).**
> It is not unique to our framework. Crucially, the computational cost of dot products is negligible compared to model inference as it involves only simple matrix multiplication after embeddings are generated.
>
> - **Our key innovation lies in accelerating the generation of embedding ($x_0$)**. As shown in Table 3, TA-Rec's inference time is lower than other diffusion recommenders and comparable to traditional recommenders like SASRec.
>
>
>
>
> ### Weakness 2
> > Random negative sampling of APA may fail to capture hard negative samples for preference alignment.
>
>
> - The negative sampling strategy and our primary innovation of the adaptive alignment coefficient $\lambda_\beta(t,d)$ are **independent effective solutions**. The coefficient $\lambda_\beta(t,d)$ dynamically adjusts the optimization strength to mitigate overfitting to ambiguous preferences and noise. **The effectiveness of APA is validated in the Sec 5.3 of our paper**. The random negative sampling we employed is indeed a simplification, and **we have acknowledged it as a limitation in Appendix E.3 of our paper**.
>
>
> - We consider your suggestion a valuable direction for future research, and we conduct additional experiments to explore hard negative sampling strategy in Table R1. The ``w/ hard`` samples 5 random negatives, selects the most similar to the target via cosine similarity as a hard negative, and conducts preference alignment without adaptive coefficients. ``ours-hard`` incorporates APA with hard negative sampling for adaptive alignment. The superiority of ``w/ hard`` over ``w/o APA`` and ``ours-hard`` over ``ours`` **demonstrates the potential of hard negative sampling**.
>
> **Table R1:**  Experiments on hard negative sampling strategy.
> | Methods | YooChoose |  | KuaiRec |  | Zhihu |  |
> | :---: | :---: | :---: | :---: | :---: | :---: | :---: |
> |  | H@20| N@20|H@20| N@20| H@20 | N@20|
> |w/o APA| 5.89|3.82| 5.33|4.30|2.41|0.89|
> |w/ hard|6.08|4.01|5.36|4.39|2.40|0.92|
> |ours|6.15|4.06|5.40|4.42|2.43|0.91|
> |ours-hard|6.16|4.08|5.42|4.40|2.45|0.93|
>
>
>
>
> ### Weakness 3
> > Performance in data-sparse or diverse scenarios
>
> - While cold-start and diversity are important challenges in recommendation systems, they are orthogonal to our core contribution: ​​resolving the fundamental efficiency-effectiveness trade-off in diffusion-based recommenders​​ deployment.
>
> - Extending our approach to more challenging scenarios, such as cold-start problems or diversity, could be a focus for future research. Here, we construct datasets in data-sparse setting for exploration, by removing 30% interactions. The experimental results are presented in Table R2. **The superior performance of our method compared to the baseline indicates its robustness in data-sparse scenarios**. Additionally, to validate the diversity of our TA-Rec, we conduct additional experiments on the coverage metrics, which measure the proportion of unique items recommended across multiple users. The experimental results presented in Table R3 indicate that both TA-Rec and DreamRec, which leverage diffusion models for generating recommendations, exhibit greater diversity compared to traditional recommenders like SASRec. **This underscores the advantages of TA-Rec in achieving diverse recommendations**.
>
>
> **Table R2:**  Experiments on data-sparse scenarios.
> | Methods | YooChoose |  | KuaiRec |  | Zhihu |  |
> | :---: | :---: | :---: | :---: | :---: | :---: | :---: |
> |  | H@20 | N@20| H@20 | N@20| H@20| N@20|
> |SASRec|2.89|1.45|3.28| 1.19|1.44|0.57|
> |DreamRec|4.29|2.15|5.02|3.96|2.15|0.77|
> |ours|5.65|3.46|5.19|4.25|2.36|0.89|
>
>
> **Table R3:**  Experiments on the diversity of diffusion recommenders.
> | Methods | YooChoose  | KuaiRec |   Zhihu |
> | :---: | :---: | :---: | :---: |
> |  | Cover@20| Cover@20| Cover@20|
> |SASRec|0.1703|0.8368|0.7306|
> |DreamRec|0.2051|0.8426|0.7616|
> |ours|0.2042|0.8416|0.7609|
>
>
>
> ### Questions:
> > Online Deployment and Inference Time & Lipschitz Continuity Assumptions & Online Learning
>
> (1) By achieving one-step generation with TCR in the pretraining stage, TA-Rec has similar inference time to traditional recommenders (e.g., SASRec), as shown in Table 3 of our paper. **Thus, our method can be effectively deployed in production environments**. (2) The Lipschitz continuity assumption is not an inherent assumption applied to data, **but rather a desirable property that we realize on the denoising function $f_\theta$ through TCR objective**. By constraining TCR loss, we enforce local smoothness between adjacent denoising steps and ensure $f_\theta$ satisfies the Lipschitz continuity assumption. Then, $f_\theta$ can achieve one-step generation with bounded error, mapping noise directly to oracle items as shown in Theorem 1.
> (3) Our algorithm primarily focuses on offline training setting, which is consistent with comparable methods like DreamRec and PreferDiff. We agree that it can support real-time incremental and streaming learning, which is a potential direction for future work.
>
>
> We sincerely hope that our response has adequately **addressed your concerns**. If so, we would greatly appreciate your consideration in **raising the score**. If there are any remaining concerns, **please let us know**, and we will **continue to actively address your comments** and **improve our work**.

---

> > ### Comment · Reviewer_BncC · 2025-08-08
> >
> > I thank the authors for their response. The rebuttal has clarified my concerns, and  I will therefore maintain my original score.

---

> > > ### Author Response · Authors · 2025-08-08
> > >
> > > Dear reviewer:
> > >
> > > Thank you for your feedback. We are pleased that our rebuttal **clarified your concerns**, and we respect your decision to maintain the score. We appreciate your time and engagement throughout this process.

---

> ### Comment · Area_Chair_hNZE · 2025-08-04
>
> Hi reviewer BncC,
> This is a gentle reminder that the authors added their response on your question.
> Can you provide your feedback on their response?
> Please keep in mind that the deadline of August 6th approaching, and your additional timely feedback greatly enhance further discussions if needed.
>
> Thanks,
> Area Chair.

---

> ### Author Response · Authors · 2025-08-07
> **Looking forward to your reply**
>
> Dear Reviewer BncC,
>
> Thank you again for your constructive comments. We have thoroughly addressed your concerns by further investigating a hard negative sampling strategy, evaluating the performance of our TA-Rec in data-sparse and diverse scenarios through additional experiments, and clarifying its efficiency and potential for online learning.
>
> We look forward to further discussion with you and would greatly appreciate your positive feedback on our rebuttal.
>
> Best regards,
>
> The Authors

---

### Note · Authors · 2025-08-13

Dear Reviewers, ACs, SACs, and PCs,

We sincerely appreciate your time and effort throughout the review, rebuttal, and discussion stages. Below, we summarize the strengths of our work as highlighted by the reviewers,  the key concerns raised, our responses addressing these concerns, and the reviewers' feedback.

**Strengths:**

* **Novelty** and **Significance:** a pioneering effort (Reviewer ``ptjv``), making a meaningful contribution (Reviewer ``J9pE``), well-motivated (Reviewer ``dntH``), and a novel approach to a significant challenge (Reviewer ``NQjp``).
* **Quality** and **Soundness:** well-written, well-structured, easy to follow (Reviewer ``ptjv``, ``dntH``, ``NQjp``), well-grounded theoretical analysis (``J9pE``, ``NQjp``, ``dntH``), extensive and convincing experiments (Reviewer ``NQjp``, ``ptjv``).

**Main concerns:**

1. Performance on **more datasets** (Reviewer ``NQjp``, ``dntH``)
2. Performance of **recommendation diversity** (Reviewer ``BncC``, ``J9pE``, ``ptjv``)
3. Further analysis of more **negative sampling strategies** (Reviewer ``BncC``, ``NQjp``)
4. The **core novelty** of the proposed method (Reviewer ``ptjv``)

**Our responses:**

1. We conducted more experiments on **three additional datasets** with consistent performance improvement over baselines.
2. We added a **diversity analysis** using the Coverage@20 metric, showing the superiority of TA-Rec in diversity.
3. We explored **alternative negative sampling strategies** (hard negative and popularity-based), demonstrating that our APA module is robust and its effectiveness is independent of the sampling method.
4. We clarified that the core novelty lies in **new problem formulation** through discretization error, a new **two-stage framework**, and an original **theoretical contribution**.

**Reviewers' feedback:**

We are grateful to reviewers who have replied (``BncC``, ``J9pE``, ``dntH``, ``ptjv``) for confirming that their main concerns were **satisfactorily addressed** and **everything is good**.
And we believe the main concerns regarding more datasets and negative sampling strategies from reviewer  ``NQjp`` (conf 2) have been addressed as well.

We promise that all additional experiments and analyses will be incorporated into the revised paper. We hope the improvements made will be taken into consideration. We sincerely appreciate your valuable time!

Best regards,

Authors of paper #16256

---

### Decision · Program_Chairs · 2025-09-17

**Decision:**

Accept (poster)

**Comment:**

This paper proposes TA-Rec, a two-stage diffusion-based recommendation framework designed to address the efficiency–effectiveness trade-off induced by discretization error. The pretraining stage introduces Temporal Consistency Regularization (TCR) to smooth the denoising trajectory and enable one-step generation, while the fine-tuning stage employs Adaptive Preference Alignment (APA) to better align generation with user preferences. Theoretical analysis provides an error bound for accelerated inference, and extensive experiments show that TA-Rec achieves efficiency comparable to transformer recommenders while surpassing multi-step diffusion baselines in accuracy.

The strengths of the paper are that it identifies a practically important problem—balancing inference efficiency and accuracy in diffusion recommenders—which had not been systematically addressed before (Reviewer ptjv, Reviewer dntH). The proposed TCR and APA modules are well-motivated, with bounded-error analysis giving theoretical grounding (Reviewer J9pE, Reviewer NQjp). Empirical validation is thorough: the method consistently outperforms diffusion and non-diffusion baselines across multiple datasets, showing 100× faster inference than DreamRec and measurable accuracy gains (Reviewer dntH). Reviewers also highlighted that the paper is well-written and easy to follow (Reviewer ptjv, Reviewer NQjp, Reviewer dntH).

The weaknesses concern novelty and scope. Reviewer ptjv argued that TCR is closely related to existing consistency models and APA to DPO, questioning whether the work represents a conceptual advance rather than an adaptation. Reviewer J9pE initially questioned whether one-step denoising undermines the essence of diffusion and whether diversity benefits are preserved; this was later addressed with diversity experiments and theoretical clarification. Reviewer NQjp and Reviewer dntH raised concerns about dataset diversity, lack of computational complexity analysis, and limited ablations on negative sampling. These points were addressed in rebuttal with new experiments on three additional datasets, coverage@20 analysis for diversity, detailed complexity derivations, and sensitivity tests with hard and popularity-based negatives. Reviewer BncC noted possible computational overhead in small-scale settings and reliance on random negative sampling, which were clarified by showing negligible cost relative to inference and by reporting robustness to alternative sampling strategies. Nonetheless, questions remain about the degree of conceptual novelty and the realism of some theoretical assumptions (e.g., pretraining loss tending to zero in Theorem 1, as noted by Reviewer ptjv).

During discussion, most reviewers acknowledged that their concerns were satisfactorily addressed. Reviewer J9pE explicitly raised their score after rebuttal. Reviewer NQjp and Reviewer dntH maintained borderline scores despite the new evidence, citing remaining reservations about evaluation scope and training overhead. Reviewer ptjv maintained a borderline accept, emphasizing limited novelty, though recognizing the empirical strength and engineering effort.

Overall, the paper makes a technically solid contribution by formulating an important problem, designing a coherent two-stage solution with both theoretical and empirical support, and showing clear efficiency–effectiveness improvements. While the degree of novelty is debated, the balance of evidence indicates that the work advances the state of diffusion-based recommendation in a meaningful way.

Final Recommendation: Accept